# Practical and Scalable Hamiltonian Monte Carlo Without the Metropolis Test

**Jakob Robnik** [1]  **Reuben Cohn-Gordon** [1]  **Uroš Seljak** [1][2]

## Abstract

Hamiltonian Monte Carlo and underdamped Langevin Monte Carlo are leading methods for sampling from high-dimensional distributions with differentiable densities. Both rely on numerical integration, which introduces asymptotic bias in expectation estimates. This bias can be removed by adjusting the numerical integration with a Metropolis–Hastings (MH) step, at a cost of slower mixing and larger variance. Alternatively, we can trade bias for lower variance if we avoid the MH step and use an appropriate step size of integration. These unadjusted schemes have strong performance, especially in high-dimensional problems, but are rarely used due to the lack of automated step-size selection. We propose an automatic tuning scheme that selects a step size to meet a user-specified asymptotic bias tolerance. The method is based on a relationship between energy error and bias which we establish. We rigorously analyze the method in the Gaussian setting and numerically extend the analysis to several non-Gaussian problems. Experiments on Bayesian inference and large-scale statistical physics models (with over one million parameters) show that, with our tuning, unadjusted methods consistently and significantly outperform adjusted counterparts.

## 1. Introduction

Sampling offers a way to compute expectation values $\mathbb{E}_p[f] = \int p(\boldsymbol{x})f(\boldsymbol{x})d\boldsymbol{x}$, where $f(\boldsymbol{x})$ is some function of parameters $\boldsymbol{x} \in \mathbb{R}^d$ and $p(\boldsymbol{x}) = e^{-\mathcal{L}(\boldsymbol{x})}/Z$ is a given probability density, often with $Z = \int e^{-\mathcal{L}(\boldsymbol{x})}d\boldsymbol{x}$ unknown. This is a key tool in many disciplines, from social science (Gelman et al., 2013), to high energy physics (Duane et al., 1987),

[1]Department of Physics, University of California, Berkeley, CA 94720, USA [2]Physics Division, Lawrence Berkeley National Laboratory, Berkeley, CA 94720, USA. Correspondence to: Jakob Robnik <jakob_robnik@berkeley.edu>.

*Proceedings of the $43^{rd}$ International Conference on Machine Learning*, Seoul, South Korea. PMLR 306, 2026. Copyright 2026 by the author(s).

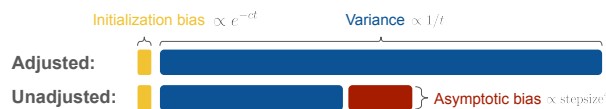

*Figure 1.* Graphical representation of the bias decomposition from Equation (2). Both adjusted and unadjusted methods have initialization bias, which typically decays exponentially fast for HMC-like algorithms (Margossian & Gelman, 2024) (with a rate denoted by $c$ in the Figure). Both methods also have the variance associated with the finite number of samples taken. It decays inversely proportionally to the number of samples $t$, with a proportionality constant that is determined by the autocorrelation time. Unadjusted methods additionally have asymptotic bias, which depends strongly on the step size $\epsilon$; as $\mathcal{O}(\epsilon^4)$, for the example from Appendix D.

computational chemistry (Leimkuhler & Matthews, 2015), statistical physics (Leimkuhler & Matthews, 2015), and machine learning (Neal, 2012). Often a gradient $\nabla\mathcal{L}(\boldsymbol{x})$ is available, either analytically, or via automatic differentiation (Griewank & Walther, 2008; Margossian, 2019); examples appear in Bayesian statistics (Štrumbelj et al., 2024; Carpenter et al., 2017), machine learning (Baydin et al., 2018), lattice quantum problems (Gattringer & Lang, 2010), and cosmology (Campagne et al., 2023; Horowitz & Lukic, 2025).

Markov Chain Monte Carlo (MCMC; (Metropolis et al., 1953)) is a commonly employed class of sampling methods in which a Markov chain $\{\boldsymbol{x}_i\}_{i=1}^n$ is designed to have a stationary distribution $p(\boldsymbol{x})$ (or close to $p$), so that the expectation value $\mathbb{E}_p[f]$ can be approximated by $\bar{f} = \frac{1}{n}\sum_{i=1}^n f(\boldsymbol{x}_i)$. When a (smooth) gradient is available, Hamiltonian Monte Carlo (HMC; Duane et al. (1987); Neal (2011); Betancourt (2018)) and underdamped Langevin Monte Carlo (LMC; Horowitz (1991); Leimkuhler & Matthews (2015)) are state-of-the-art algorithms. Despite their success (Štrumbelj et al., 2024), MCMC often presents a computational bottleneck (Gattringer & Lang, 2010; Leimkuhler & Matthews, 2015; Simon-Onfroy et al., 2025), especially in the high-dimensional applications, forcing practitioners to resort to more approximate methods, such as the Laplace approximation (Millea & Seljak, 2022) or an ensemble Kalman filter (Houtekamer & Mitchell, 2005).

**Bias-variance tradeoff** To understand the challenge faced by MCMC, consider the root mean squared error (RMSE) of the Monte Carlo estimator:

$$\text{RMSE}[\bar{f}] = \mathbb{E}_{MC}[(\bar{f} - \mathbb{E}_p[f])^2]^{1/2}, \quad (1)$$

where $\mathbb{E}_{MC}[\cdot]$ denotes the expectation with respect to the initialization and Markov transition randomness. The RMSE can be decomposed as

$$\text{RMSE}[\bar{f}]^2 = \text{Bias}[\bar{f}]^2 + \text{Var}[\bar{f}], \quad (2)$$

where $\text{Bias}[\bar{f}] = \mathbb{E}_{MC}[\bar{f} - \mathbb{E}_p[f]]$ and $\text{Var}[\bar{f}] = \mathbb{E}_{MC}[(\bar{f} - \mathbb{E}_{MC}[\bar{f}])^2]$. The bias term arises either from the chain not yet reaching its stationary state and being influenced by its initial position (initialization bias), or because the stationary distribution of the chain $\tilde{p}(\boldsymbol{x})$ does not equal the target distribution $p(\boldsymbol{x})$ (asymptotic bias). Initialization bias is an important issue (Margossian et al., 2024), but we will not study it here, noting that for long enough chains it can be eliminated by discarding initial samples (Margossian & Gelman, 2024). The variance term comes from the finite chain length and the correlation between the samples. This decomposition is illustrated in Figure 1.

One of the main practical goals in designing MCMC methods is to obtain a desired RMSE at the lowest possible computational budget. This amounts to balancing the cost of generating samples, correlations between the samples, and the asymptotic bias. For example, the use of Metropolis-Hastings adjustment (Chib & Greenberg, 1995) ensures that a chain satisfies detailed balance, so that the asymptotic bias vanishes, but for finite-length chains, it does not remove the variance. Noting that chains are finite in practice, one could negotiate the tradeoff differently and reduce the variance at the cost of introducing some bias. This strategy is the focus of the present work.

**Illustrative example** The shortcoming of performing MH is the scaling with the dimension that it implies for the sampler. To develop an intuition, we consider a simple problem, where the target is a product of $K$ independent $D$-dimensional distributions $q$:

$$p(\boldsymbol{x}) = \prod_{i=1}^{K} q(x_{iD}, x_{iD+1} \ldots x_{iD+D-1}),$$

where $D$ is a small number. Suppose we are interested in the expectation value of a function of only one of those parameters, such as $f(\boldsymbol{x}) = x_1^2$. We will measure the performance as the number of evaluations of $\nabla p(\boldsymbol{x})$ (which is typically the bottleneck of computation and thus a proxy for the wall-clock time) that the sampler uses to get the relative RMSE, i.e. $\text{RMSE}[\bar{f}]/\mathbb{E}_p[f]$ (averaged over 128 chains) below 10%.

Figure 2 shows performance as a function of the dimension of the problem $d = KD$. As can be seen, the number of gradient calls scales as $d^{1/4}$ for Metropolis adjusted methods, in accordance with (Beskos et al., 2013). This is because even though the Hamiltonian and Langevin dynamics operate independently on each copy of the distribution, the energy change $\Delta_H$ (defined below), and therefore the MH acceptance probability, involves a sum over all parameters. Optimal performance requires a fixed acceptance rate (Beskos et al., 2013; Neal, 2011), so to compensate for the increasing number of parameters, the step size needs to decrease, so that the integrator uses more gradient evaluations for a fixed trajectory length. Unadjusted methods, by contrast, do not suffer from this scaling since they operate on each copy independently. They need to ensure that the bias is small, but this is independent of the number of parameters in these examples.

The example in Figure 2 is idealized, but a similar scaling has been observed for mean field models (Durmus & Eberle, 2023) and real problems, like cosmological field level inference (Simon-Onfroy et al., 2025). Thus, particularly for high-dimensional problems, unadjusted samplers may be significantly more efficient. Crucially, however, they are only viable if the step size can be chosen so that the asymptotic bias is small compared to the variance of the finite set of samples taken.

**Our contributions** We propose an automated step size tuning scheme for unadjusted HMC and LMC, making them usable in a black-box manner, i.e. *without manually tuning any hyperparameters*. We expect this to have a significant practical impact on applications that require gradient-based sampling in high dimensions. The central idea is that the energy error that arises from integrating with step size $\epsilon$ provides a measure of the asymptotic bias, and step size can be adaptively varied in a short tuning phase to target an energy error which results in low asymptotic bias relative to the variance. In Section 4 we show analytically for Gaussians that the energy error variance can be used to upper bound the bias. In Section 5, we numerically confirm the analytical results and show that the same upper bound generally extends to non-Gaussian targets. In Section 6 we use the bound to construct our tuning algorithm. In Section 7 we demonstrate its effectiveness on a range of standard benchmarks and on a real-world lattice field theory problem in more than 1 million dimensions.

## 2. Related Work

HMC is a state-of-the-art Markov kernel for densities with smooth gradients, as is *underdamped* Langevin Monte Carlo (LMC), also known as the generalized HMC. LMC should be distinguished from *overdamped* Langevin dynamics and the corresponding Metropolis Adjusted Langevin Algorithm

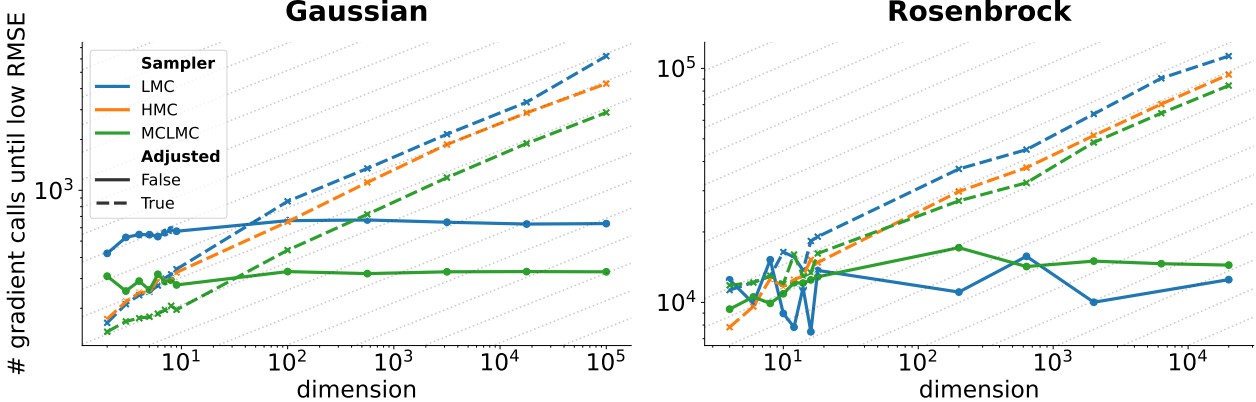

*Figure 2.* Sampling cost scaling with the dimensionality for product targets. Cost is measured by the number of gradient calls needed to achieve low error. MH adjusted methods are shown in dashed lines, and unadjusted methods are shown in solid lines. Step size for the unadjusted schemes is selected by the scheme proposed in this paper. $d^{1/4}$ power law grey lines are shown in the background. A Standard Gaussian target is shown on the left ($D = 1$), and a product of banana-shaped Rosenbrock distributions ($D = 2$) on the right. In both cases, the cost of the MH adjusted methods scales as $d^{1/4}$, while the cost of the unadjusted method remains constant with dimension.

(MALA; Rossky et al. (1978)). In HMC and LMC, each parameter $x_i$ has an associated momentum variable $u_i$. The state of the Markov chain $\boldsymbol{x}_n$ is used as an initial condition $\boldsymbol{x}(t = 0) = \boldsymbol{x}_n$ of the dynamics, along with random initial values for the momenta $u_i(0) \sim \mathcal{N}(0, 1)$. The next state in HMC is then generated by solving for $\boldsymbol{x}(t = T)$, where $T$ is a predefined trajectory length and $\boldsymbol{z}(t) \equiv (\boldsymbol{x}(t), \boldsymbol{u}(t))$ are solutions of the Hamiltonian equations with Hamiltonian function $\mathcal{H}(\boldsymbol{z}) = \frac{1}{2} \|\boldsymbol{u}\|^2 + \mathcal{L}(\boldsymbol{x})$:

$$\frac{d}{dt}\boldsymbol{x}(t) = \boldsymbol{u}(t) \qquad \frac{d}{dt}\boldsymbol{u}(t) = -\nabla\mathcal{L}(\boldsymbol{x}). \quad (3)$$

Note that the value of the Hamiltonian function, also called the energy, is a conserved quantity of the Hamiltonian equations, meaning that the energy difference $\Delta_H(\boldsymbol{z}', \boldsymbol{z}) = H(\boldsymbol{z}') - H(\boldsymbol{z})$ vanishes for exact solutions of the Hamiltonian dynamics: $\Delta_H(\boldsymbol{z}(t), \boldsymbol{z}(0)) = 0$. In LMC, after every step with Hamiltonian dynamics, the velocity is partially randomized, which corresponds to the discretization of the Langevin stochastic differential equation (Leimkuhler & Matthews, 2015).

In statistics, MH adjusted versions of these dynamics are used almost exclusively. However, unadjusted versions are used in some fields with high-dimensional distributions, such as Lattice quantum chromodynamics (Lüscher, 2018; Clark & Kennedy, 2007) and Molecular Dynamics (Leimkuhler & Matthews, 2015). Commonly, underdamped Langevin dynamics or Nosé-Hoover thermostat (Evans & Holian, 1985) are employed. Another promising option is Microcanonical Langevin Monte Carlo (MCLMC; Robnik et al. (2024); Robnik & Seljak (2024); Minary et al. (2003); Steeg & Galstyan (2021)), which makes use of velocity norm preserving dynamics. Domain knowledge (for exam-

ple the relevant time-scales of the molecular bonds) and trial runs are used in these fields to select an appropriate step size. Sometimes a diagnostic that energy of the system should not show long term drifts is also used (Tuckerman, 2023).

Unadjusted methods have also been analyzed theoretically, establishing a bound on the asymptotic bias (Durmus & Eberle, 2023) and the mixing time (Bou-Rabee & Eberle, 2023; Camrud et al., 2024). For mean-field models, the bound on both is dimension-free (Bou-Rabee & Schuh, 2023) and thus unadjusted methods provably outperform adjusted methods. However, these bounds assume some global knowledge of the distribution that is not available in practice. A notable gap in the above work is an algorithm for choosing the step size $\epsilon$. Without a principled way to do so, general practitioners cannot use unadjusted algorithms in a black-box fashion, which gravely limits their applicability.

## 3. Measuring Asymptotic Bias

**Origin of bias** Generically, Hamiltonian equations like (3) cannot be solved exactly, so a numerical integrator like velocity Verlet (Leimkuhler & Matthews, 2015) is used to approximate it. Velocity Verlet is an example of a splitting method, where to solve the dynamics numerically, one first analytically solves for $\boldsymbol{x}$ at fixed $\boldsymbol{u}$ and vice versa. The first solution is called the position update and is given by $\Phi_\epsilon^T(\boldsymbol{z}) = (\boldsymbol{x} + \epsilon\boldsymbol{u}, \boldsymbol{u})$ for Equation (3), while the second is called the velocity update and is given by $\Phi_\epsilon^V(\boldsymbol{z}) = (\boldsymbol{x}, \boldsymbol{u} - \epsilon\nabla\mathcal{L}(\boldsymbol{x}))$. The joint solution is then approximated by a composition of these maps, which for velocity Verlet is,

$$\boldsymbol{z}(t + \epsilon) \approx \Phi_\epsilon(\boldsymbol{z}(t)) = (\Phi_{\epsilon/2}^V \circ \Phi_\epsilon^T \circ \Phi_{\epsilon/2}^V)(\boldsymbol{z}(t)). \quad (4)$$

Due to this approximation, the stationary distribution $\tilde{p}(\boldsymbol{x})$ of the sampler no longer equals the desired target distribution $p(\boldsymbol{x})$ and expectation values acquire an asymptotic bias, which vanishes in the limit of step size going to zero (Durmus & Eberle, 2023). In the case of LMC, the update is additionally complemented by partial refreshments of the velocity (Leimkuhler & Matthews, 2015):

$$\boldsymbol{z}(t+\epsilon) \approx (\Phi^O_{\epsilon/2} \circ \Phi_\epsilon \circ \Phi^O_{\epsilon/2})(\boldsymbol{z}(t)), \qquad (5)$$

where $\Phi^O_\epsilon(\boldsymbol{z}) = (\boldsymbol{x},\, e^{-\epsilon/L}\boldsymbol{u} + (\sqrt{1 - e^{-2\epsilon/L}})\mathbf{n})$ and $\mathbf{n} \sim \mathcal{N}(0, I)$. The parameter $L$ determines the amount of momentum refreshment and plays a similar role as the trajectory length in HMC.

**Summary statistics** When minimizing bias, it is important to determine the expectation with respect to which the bias is defined. For instance, an important expectation in Bayesian statistics (where quantification of uncertainty is key) is of the second moments, so that we are concerned with expectations of the form $\mathbb{E}_p[x_i x_j]$, or more generally the covariance matrix $\Sigma_p = \mathbb{E}_q[(\boldsymbol{x} - \mathbb{E}[\boldsymbol{x}])(\boldsymbol{x} - \mathbb{E}[\boldsymbol{x}])^T]$. For Gaussian distributions, the covariance matrix contains all the information about the posterior, but even for non-Gaussian distributions, they are often used as a summary statistics.

From a theoretical perspective, it may also be interesting to consider the bias of a wider set of functions, such as any Lipschitz-continuous functions. In what follows, we consider a metric that controls the more general bias and a metric measuring the covariance matrix bias.

**Covariance matrix error** To quantify expectation value error of the second moments, we introduce a scalar measure of the covariance matrix error, which we define as

$$b^2_{cov}(\Sigma_p, \Sigma_q) \equiv \frac{1}{d}\,\mathrm{Tr}\big\{(I - \Sigma_p^{-1}\Sigma_q)^2\big\}, \qquad (6)$$

where $\Sigma_p$ is the true covariance matrix of the target distribution $p$ and $\Sigma_q$ is the covariance matrix of some other distribution $q$. In the simple case where the covariance matrices are diagonal, $b_{cov}$ is the relative error of the variance estimate, averaged over the parameters, i.e., $b^2_{cov}(\Sigma_p, \Sigma_q) = \frac{1}{d}\sum_{i=1}^d ([\Sigma_p]_{ii} - [\Sigma_q]_{ii})^2/[\Sigma_p]^2_{ii}$. This convergence metric is often used in practice (Grumitt et al., 2022; Robnik et al., 2024). The diagonal form is preferable in high dimensions as it does not require storage of the full covariance matrix. Nonetheless, we will measure error with (6) when feasible because it additionally penalizes the off-diagonal terms and has a number of nice properties: it is a divergence on the space of positive-definite matrices (meaning that it is non-negative, and zero if and only if the two matrices are the same), can be connected to the effecive sample size and is invariant to the linear change of basis of the parameter space. Proofs of these properties are provided in Appendix A.

**Wasserstein distance** The Wasserstein distance $\mathcal{W}_\nu(p, q)$ between densities $p$ and $q$ is (Kantorovich, 1960):

$$\mathcal{W}_\nu(p, q) = \big(\inf_{\pi \in \Pi(p,q)} \int \|\boldsymbol{x} - \boldsymbol{x}'\|^\nu \pi(\boldsymbol{x}, \boldsymbol{x}')d\boldsymbol{x}d\boldsymbol{x}'\big)^{1/\nu}, \qquad (7)$$

where $\Pi(p, q)$ is the set of probability densities on $\mathbb{R}^d \times \mathbb{R}^d$ with marginals $p$ and $q$. Wasserstein distance has several nice properties: it is invariant to change of basis, is a metric on the space of distributions, and most importantly in the present context, it upper bounds the bias of Lipschitz-continuous functions. That is, for any Lipschitz-continuous function $f$, with Lipschitz constant $L$ (meaning that $|f(\boldsymbol{x}) - f(\boldsymbol{x}')| < L\|\boldsymbol{x} - \boldsymbol{x}'\|$ for any $\boldsymbol{x}, \boldsymbol{x}' \in \mathbb{R}^d$), the bias associated with this function is upper bounded by $\mathcal{W}_1(p, \tilde{p})$, which is in turn upper bounded by $\mathcal{W}_2(p, \tilde{p})$:

$$\mathbb{E}_p[f] - \mathbb{E}_{\tilde{p}}[f] \le L\mathcal{W}_1(p, \tilde{p}) \le L\mathcal{W}_2(p, \tilde{p}). \qquad (8)$$

The first inequality is Kantorovich-Rubinstein duality (Villani, 2003), and the second is a consequence of Jensen's inequality. We will focus on $\mathcal{W}_2$.

**Energy error variance per dimension** Our goal is to control some measure of the bias. We will focus on $b^2_{cov}$, because it is of practical interest in various fields like Bayesian inference and on $\mathcal{W}_2$, because it provides a bound on the bias of all Lipschitz continuous functions. We will do this by monitoring the energy error $\Delta_H(\Phi_\epsilon(\boldsymbol{z}(t)), \boldsymbol{z}(t))$ induced by numerical integration with step size $\epsilon$. Computing the energy error is a side product of the integration step, for example for HMC, the position update energy change is $\mathcal{L}(\boldsymbol{x} + \epsilon\boldsymbol{u}) - \mathcal{L}(\boldsymbol{x})$ and $\frac{1}{2}\|\boldsymbol{u} - \epsilon\nabla\mathcal{L}(\boldsymbol{x})\|^2 - \frac{1}{2}\|\boldsymbol{u}\|^2$ for the velocity update. For all models of practical interest this incurs a negligible cost compared to the gradient evaluation, because $\mathcal{L}(\boldsymbol{x})$ is a side product of the gradient evaluation.

We define the Energy Error Variance Per Dimension (EEVPD),

$$\mathrm{EEVPD} = \mathrm{Var}_{\boldsymbol{x} \sim \tilde{p}, \boldsymbol{u} \sim N(0, I)}[\Delta_H(\Phi_\epsilon(\boldsymbol{x}, \boldsymbol{u}), (\boldsymbol{x}, \boldsymbol{u}))]/d. \qquad (9)$$

Crucially, this is a quantity that can easily be estimated in practice: $\boldsymbol{x} \sim \tilde{p}, \boldsymbol{u} \sim \mathcal{N}(0, I)$ is the stationary distribution of the chain, so computing EEVPD amounts to collecting the samples from the stationary chain, evaluating the one-step energy error for each of those samples and computing their variance. This can be done online, using a running average of the first and second moment, so that the step size $\epsilon$ can be adaptively varied to target a desired value of EEVPD.

## 4. Analytic Results for Gaussian Distributions

We begin by showing that EEVPD can be used to control the asymptotic covariance matrix bias $b^2_{cov}(\Sigma, \tilde{\Sigma})$ and Wasserstein distance $\mathcal{W}_2(p, \tilde{p})$ for Gaussian target distributions,

$p = \mathcal{N}(0, \Sigma)$. The key tool is that the stationary distribution $\tilde{p}$ of unadjusted HMC and LMC for Gaussians is known exactly (Gouraud et al., 2025) and is also Gaussian, with $\tilde{p} = \mathcal{N}(0, \tilde{\Sigma})$. $\tilde{\Sigma}$ has the same eigenvectors as $\Sigma$, but its eigenvalue associated to the $i$-th eigenvector is

$$\tilde{\sigma}_i^2 = \frac{\sigma_i^2}{1 - \epsilon^2/4\sigma_i^2}, \tag{10}$$

where $\sigma_i^2$ is the corresponding eigenvalue of $\Sigma$. Therefore the EEVPD has a closed form:

**Lemma 4.1.** *For a Gaussian distribution with covariance matrix eigenvalues $\{\sigma_i^2\}_{i=1}^d$ and an HMC or LMC sampler with a stable velocity Verlet integrator, meaning that step size $\epsilon < 2\min_i \sigma_i$,*

$$\mathrm{EEVPD} = \frac{1}{d}\sum_{i=1}^d E(\epsilon^2/\sigma_i^2),$$

*where $E(y) = \frac{y^3}{16(1-y/4)}$.*

The key result of this section is then:

**Theorem 4.2.** *For a Gaussian distribution $\mathcal{N}(0, \Sigma)$ with covariance matrix eigenvalues $\{\sigma_i^2\}_{i=1}^d$ and HMC or LMC sampler with a stable velocity Verlet integrator, meaning that the step size $\epsilon < 2\min_i \sigma_i$, the covariance matrix bias is upper bounded by*

$$b_{cov}^2(\Sigma, \tilde{\Sigma}) \leq \varphi^{-1}(\mathrm{EEVPD}),$$

*as long as* $\mathrm{EEVPD} < 0.397$. *Here,*

$$\varphi(x) = \frac{4x^{3/2}}{(1 + x^{1/2})^2}.$$

*Similarly, the Wasserstein distance between the target and stationary distributions is upper bounded by*

$$\mathcal{W}_2(p, \tilde{p})^2/d \leq \epsilon^2 \varphi_W^{-1}(\mathrm{EEVPD}),$$

*as long as* $\mathrm{EEVPD} < 6.75$. *Here, $\varphi_W \equiv E \circ W^{-1}$ with $W(y) = \frac{2(1-y/8-\sqrt{1-y/4})}{y(1-y/4)}$. Both bounds are sharp and realized if and only if the target is isotropic, i.e. $\Sigma_p \propto I$.*

All the proofs are given in Appendix B. Note that the conditions on EEVPD are not a severe limitation in practice, see for example Table 2, where significantly lower values of EEVPD are used.

# 5. Numeric Results for Non-Gaussian Distributions

We now examine the validity of Theorem 4.2 for standard Bayesian inference benchmark problems. We focus on verifying the bound on $b_{\mathrm{cov}}$, due to the difficulty of numerically

computing the Wasserstein distance for the problems considered here. Even though the exact asymptotic error-EEVPD relation of Theorem 4.2 applies only for HMC and LMC, unadjusted MCLMC has a notion of energy (Robnik et al., 2024), so we will also test MCLMC here. Benchmark problem descriptions are in Appendix F.1. For each problem, we show the asymptotic value of $b_{cov}(\Sigma, \tilde{\Sigma})^2$ as a function of EEVPD in Figure 3. Asymptotic $b_{cov}$ is computed by running unadjusted chains with different step sizes, each using $10^8$ gradient calls. We eliminate the initial $10^4$ calls to eliminate the initialization bias and use the subsequent samples to compute the expectation values for the covariance matrix and EEVPD. We monitor $b_{cov}(\Sigma, \tilde{\Sigma})^2$ from Equation (6) and check that it has converged to the asymptotic value. If the convergence has not yet been achieved, i.e., the bias is still decaying, we do not show these measurements on the plots. This happens for some of the harder problems at small step sizes, where the chains are not long enough for the variance to become negligible. We have checked that the variation between the chains is negligible, but nonetheless average the results over 4 independent chains.

Numerical results for unadjusted HMC on Gaussians agree perfectly with Theorem 4.2: for the standard Gaussian, the equality holds, while for the Ill-conditioned Gaussian, the inequality holds for EEVPD $< 0.397$ as per the theorem. The inequality also applies to the majority of non-Gaussian benchmark problems, illustrating its broader applicability. One exception is the Brownian motion example, where it is off by approximately a factor of $1.5$ at small $\epsilon$, meaning that one would think one has $< 2\%$ asymptotic error, when in fact it was $3\%$. The Rosenbrock and Funnel examples are only shown at small step sizes, because the problem becomes numerically unstable at higher step sizes, incurring divergences. Very similar results also apply to MCLMC, except that the bias at a fixed EEVPD is usually lower for MCLMC. This suggests that the tuning scheme we develop for unadjusted HMC can also be applied to MCLMC, but a slightly larger EEVPD may be used.

# 6. Automatic Tuning Scheme

We now present a tuning scheme that automatically determines the stepsize, trajectory length and a preconditioner of the unadjusted methods. After the tuning phase is finished, the hyperparameters are frozen and the sampling begins.

## 6.1. Stepsize

The core findings of sections 4 and 5 are that by controlling EEVPD, we can in turn control the asymptotic bias of expectations of interest. Our tuning scheme for step size is therefore straightforward: for any unadjusted sampler with an appropriate notion of energy, we keep a running estimate of EEVPD, and adaptively vary $\epsilon$ to target a desired value

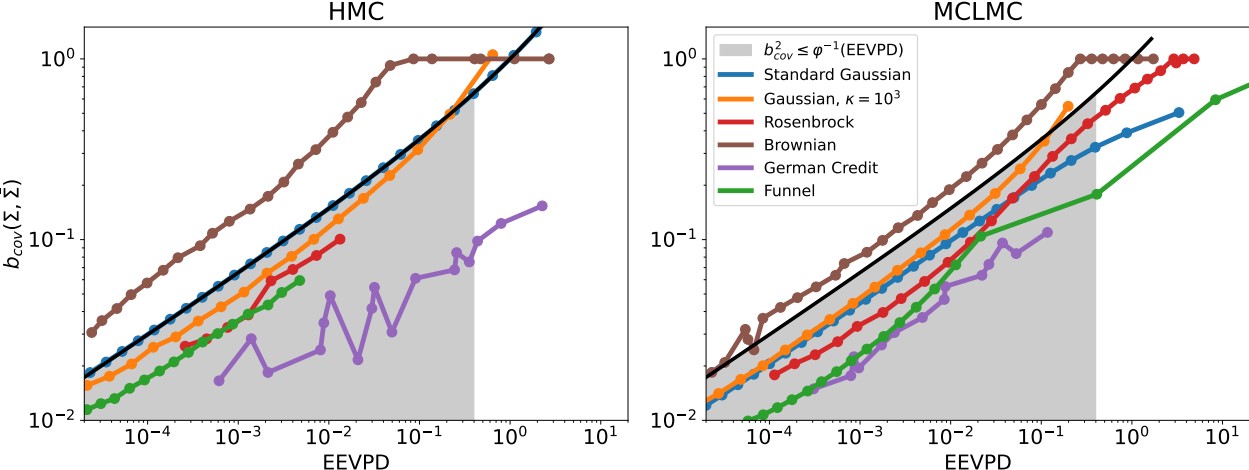

*Figure 3.* The asymptotic covariance matrix error $b_{cov}(\Sigma, \tilde{\Sigma})$ as a function of EEVPD. Unadjusted HMC (left), and unadjusted MCLMC (right) are shown. The relation is shown for various problems from Section 5. The analytical equality for the Standard Gaussians from Theorem 4.2 is shown in black and agrees perfectly with the numerical results for HMC (left). The inequality for arbitrary Gaussian distributions is shown as a shaded grey region, and also applies perfectly for Gaussians up to EEVPD = 0.397, as per Theorem 4.2. We see that most targets abide by this inequality.

of EEVPD in a stochastic optimization scheme. In practice, an optimization algorithm such as dual averaging (Nesterov, 2009; Hoffman & Gelman, 2014) can be used here, but the choice of optimizer has little effect on the results; we use the scheme in Appendix E.

Note that this approach is similar to how the step size is adapted in Metropolis adjusted methods where an acceptance rate is targeted instead of the EEVPD. Both quantities are based on energy error. However, note that the EEVPD value that we target does not correspond to the acceptance rate values that would be targeted in Metropolis adjusted methods. For example, if one would simply determine the stepsize based on the acceptance rate and then skip the MH step, the resulting method would have a nonzero bias, which would however typically (in high dimensions) be much smaller than the variance, making it suboptimal.

**Speed of convergence**  A running average estimate of EEVPD converges significantly faster than say, a running average of the second moments. This is because the energy changes in the subsequent steps are very mildly correlated. For example, integrated autocorrelation time for EEVPD time series with MCLMC dynamics is around 2 for standard Gaussian problem in $d = 100$ and around 5 for the Brownian motion problem from Appendix F. Therefore, the variance of the EEVPD estimate is small, even for a relatively small number of samples. We observe that for the experiments considered in this work, the EEVPD converges to the desired value in an order of a few tens of steps. This is analogous to how the acceptance rate is cheap to estimate.

Note that the EEVPD tuning needs to run for long enough that the initialization bias becomes negligible, just as the acceptance rate tuning in the adjusted algorithms. The step size is continuously adapted throughout the burn-in such that the initial samples are quickly forgotten. The rate of forgetting is set by the parameter gamma (see Appendix E) which we fix to ensure a decay time of 50 steps. This is typically short compared to the length of the burn-in, thus the initial samples do not affect the final step size.

**Target EEVPD**  It remains to select the desired bias tolerance and the corresponding desired EEVPD. This choice depends on the application and the accuracy requirements; we, nonetheless, provide some guidance. It is clear that the asymptotic bias squared should be smaller than the mean squared error tolerance (because mean squared error is composed of the bias and the variance, which are both non-negative), but by how much? Appendix D shows that for estimating $\mathbb{E}[x^2]$ of a standard normal distribution with finite chain length of unadjusted HMC, the optimal squared bias should be one fifth of the mean squared error tolerance. This suggests that EEVPD of $3 \times 10^{-4}$ should be used if 10% relative RMSE is required and $3 \times 10^{-7}$ if relative RMSE 1% is required. For convenience, a conversion table 2 is provided in appendix.

### 6.2. Trajectory length and diagonal preconditioning

The second hyperparameter of unadjusted HMC is the number of steps between momentum refreshes, and for unadjusted LMC, the amount of noise in each partial refresh.

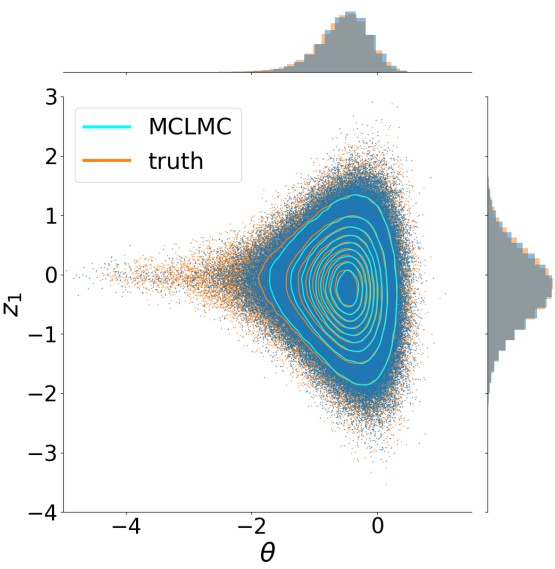

*Figure 4.* Posterior density for the funnel problem. The 2D marginal distribution in the $\theta - z_1$ plane and the corresponding 1D marginals are shown. The contours are obtained by kernel density estimation, and samples are shown as dots. The ground truth, obtained by a very long NUTS chain, is shown in orange, and unadjusted MCLMC is shown in blue. Both chains are $10^7$ samples long to eliminate the variance error. The two methods give practically indistinguishable posteriors, demonstrating that the discretization bias was successfully suppressed by the EEVPD control.

Both can be understood in terms of a single hyperparameter, the *momentum decoherence length $L$*. For HMC, $L$ is simply the number of steps in a trajectory times the step size, since this is precisely the length along the trajectory after which momentum completely decoheres. For LMC, $L$ is given in Section 3. In either case, we select $L$ based on the autocorrelation length of the the chain, as in (Robnik et al., 2024; 2025). Finally, we do a short prerun to find a diagonal preconditioning matrix, as in (Robnik et al., 2024). We find that the preconditioning matrix converges much faster with the unadjusted methods and therefore recommend using them for preconditioning, even when adjusted methods are later used for sampling. This is because the bias in the preconditioner is acceptable even if one desires asymptotically unbiased samples.

# 7. Experiments

Our central contribution is a scheme which makes unadjusted HMC, LMC and MCLMC *black-box samplers*, in the sense of requiring no manual tuning on the part of a user. It is therefore natural to compare performance to the state of the art black-box sampler, the No-U-Turn Sampler (NUTS; Hoffman & Gelman (2014)). In addition, it is of interest to compare unadjusted samplers to their adjusted counterparts.

**Samplers** With this in mind, we report results on NUTS, unadjusted LMC (uLMC), adjusted LMC (aLMC), in particular, the version proposed in (Riou-Durand & Vogrinc, 2023), which is also known as Metropolis Adjusted Langevin Trajectories (MALT), unadjusted MCLMC (uMCLMC), and adjusted MCLMC (aMCLMC). We omit reporting of unadjusted and adjusted HMC, since the performance and implementation closely resemble that of LMC, and LMC is generally considered the preferred option (Riou-Durand & Vogrinc, 2023).

**Tuning** In all cases we determine the step size by running a short warm-up chain. We take these steps as a burn-in, and initialize the chain with the final state returned by the tuning procedure. For tuning the unadjusted methods, we use the algorithm from Appendix E and target EEVPD of $3 \times 10^{-4}$, corresponding to the RMSE = 10%, which will be our notion of convergence. For MCLMC we use a slightly larger value of $5 \times 10^{-4}$, as suggested by Figure 3. In Appendix G.2, we perform an ablation study for LMC, to examine the change in performance as desired EEVPD is varied. The performance does not change much in a range of reasonable EEVPD, and the value of $3 \times 10^{-4}$ is conservative, in the sense that larger values improve performance. The only exception is Stochastic Volatility, where we find that a smaller value is needed; we use $5 \times 10^{-7}$ for HMC, and $2 \times 10^{-8}$ for MCLMC. For adjusted MCLMC we use the dual averaging algorithm (Nesterov, 2009) from (Hoffman & Gelman, 2014) to tune the step size to achieve an acceptance rate of 90%. NUTS is run with the BlackJax (Cabezas et al., 2024) window adaptation scheme. While the other algorithms are evaluated with their respective tuning schemes, for aLMC, we perform a grid search to demonstrate that black-box uLMC outperforms even optimal aLMC. We perform a search over different values of trajectory length, and at each, choose $\epsilon$ to target an acceptance rate of 80%.

**Evaluation metric** In addition to $b_{cov}^2(\Sigma, \Sigma_{\text{sampler}})$ we consider a more standard metric of convergence: following (Hoffman & Sountsov, 2022) we define the squared error of the expectation value $\mathbb{E}[f(\boldsymbol{x})]$ as

$$b^2(f) = \frac{(\mathbb{E}_{\text{sampler}}[f] - \mathbb{E}[f])^2}{\text{Var}[f]}, \qquad (11)$$

and consider the average second-moment error across parameters, $b_{avg}^2 \equiv d^{-1} \sum_{i=1}^{d} b^2(x_i^2)$. $b_{avg}^2$ can be interpreted as the accuracy equivalent to 100 effective samples (Hoffman & Sountsov, 2022). In typical applications, computing the gradients $\nabla \log p(\boldsymbol{x})$ dominates the total sampling cost, so we take the number of gradient evaluations as a proxy of a wall-clock time. As in (Hoffman & Sountsov, 2022), we measure the sampler's performance as the number of gradient calls $n$ needed to achieve low error, $b_{avg}^2 < 0.01$ or

| | aLMC
*grid search* | uLMC
*EEVPD* | aMCLMC
*acc. rate* | uMCLMC
*EEVPD* | uLMC
*grid search* | NUTS
*acc. rate* |
|---|---|---|---|---|---|---|
| Standard Gaussian | 803 | **563** | 436 | **246** | 568 | 2391 |
| Rosenbrock | 19,862 | **16,820** | 18,214 | **10,688** | 8410 | 27,070 |
| Brownian | 4667 | **2168** | 2876 | **1628** | 2407 | 5334 |
| German Credit | 7924 | **4730** | 6123 | **3960** | 4423 | 10,484 |
| Item Response | 5234 | **2020** | 5470 | **1612** | 930 | 6944 |
| Stochastic Volatility | **37,904** | 40,131 | 38,357 | **16,854** | 26,982 | 30,234 |

*Table 1.* Number of gradient calls needed to get $b_{avg}^2$ below 0.01. Lower is better. Stepsize tuning method is denoted for each sampler.

$b_{cov}^2 < 0.01$. We avoid using the effective sample size from the chain autocorrelation (Gelman et al., 2013) because it would give unadjusted methods an unfair advantage, given that it only measures the Monte Carlo variance error, but not the asymptotic bias.

For each model we run at least 128 chains, and take the median of the error across chains at each step to reduce the uncertainty. We estimate the uncertainty of results by a bootstrapping procedure (see Table 4).

**Bayesian benchmarks** Table 1 shows the results on a set of common benchmarks for Bayesian inference adapted from the Inference Gym (Sountsov et al., 2020) and described in appendix F.1. The corresponding results for the $b_{cov}$ metric are shown in Table 3 in the appendix. The problems vary in dimensionality (36–2429) and are both synthetic (the first three problems) and with real data (the last three problems). We see that the unadjusted algorithms almost always perform better than their adjusted counterparts and better than NUTS. This is especially impressive since the hyperparameters of the adjusted samplers were found by grid search, or were shown to be near optimal in the case of aMCLMC (Robnik et al., 2025). Brownian motion was the only problem where EEVPD-based bound in Section 5 has failed, but here the unadjusted samplers nonetheless converge to the desired error and achieve close to optimal performance. Tables 1 and 3 also show unadjusted LMC with hyperparameters $(L, \epsilon)$ determined by grid search. This shows that the performance of LMC with our tuning is close to optimal, except for Rosenbrock and Item response where performance is off by a factor of two, due to our conservative choice of EEVPD (see Appendix G.2).

**Marginal posterior** Figure 4 shows the marginal posterior density for the funnel problem, obtained by the unadjusted MCLMC, requiring the asymptotic bias of 1%. The posterior is practically indistinguishable from a very long NUTS chain, demonstrating that our scheme produces accurate marginal posteriors with negligible discretization bias.

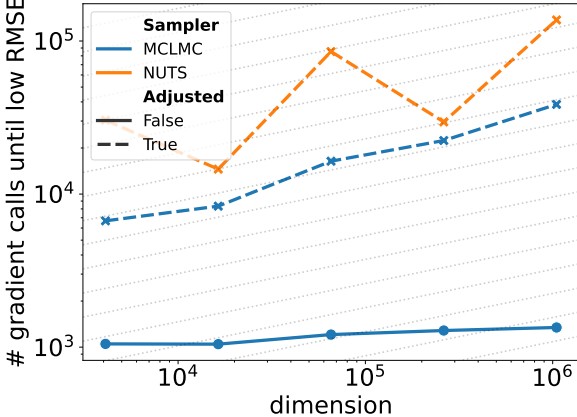

*Figure 5.* Performance on the $\phi^4$ model as lattice size is increased. Cost of the Metropolis adjusted MCLMC and NUTS grows with the number of parameters as $d^{1/4}$ (Grey lines in the background), while unadjusted MCLMC performance stays constant.

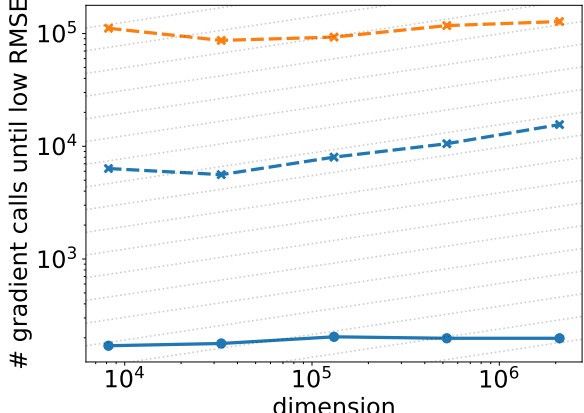

*Figure 6.* Performance on the U(1) lattice gauge theory model as lattice size is increased. Cost of the Metropolis adjusted MCLMC grows with the number of parameters as $d^{1/4}$ (Grey lines in the background), while unadjusted MCLMC performance stays constant. The legend is as in Figure 5.

$\phi^4$ **field theory**  We now apply our scheme to a real-world problem from statistical physics, the $\phi^4$ model in $1024^2 > 10^6$ dimensions (see Appendix F.2). We focus on the unadjusted microcanonical sampler since it dominated in all the above experiments and compare it to the NUTS baseline. We find that NUTS obtains $b_{\mathrm{avg}}^2 < 0.01$ in $133,266$ gradient calls. Adjusted MCLMC yields the same in $39,240$ calls, while unadjusted MCLMC remarkably only needs 1344 gradient calls, a 100 fold improvement over NUTS. Figure 5 shows that this is as an exemplary case of how the unadjusted methods achieve a constant efficiency with varying dimensionality, here on a non-product, realistic problem.

**U(1) gauge field theory**  We test our scheme on a quantum field theory problem, the U(1) model in up to two million dimensions (for the model details see Appendix F.3). We find that unadjusted schemes drastically outperform their adjusted counterparts: at two million dimensions the adjusted NUTS, HMC and MCLMC converge in 127,875, 17,699 and 15,578 gradients respectively, while unadjusted MCLMC converges in only 198 gradients. Figure 6 again shows the flat dimensionality scaling of efficiency for unadjusted schemes.

## 8. Limitations

The proposed scheme automatically selects a step size that in general performs well, both for Gaussian distributions and for a variety of non-Gaussian distributions with shapes ranging from a banana (Rosenbrock function) to the funnel structure (see Figure 4) and Mexican hat type potentials ($\phi^4$ field theory). However, we have identified a model (Brownian motion) where the bias is larger than expected. Theoretical results for Gaussians suggest that this cannot be due to ill-conditionedness, if anything, ill-conditionedness should reduce the bias as non-standard Gaussian distributions have smaller bias (at fixed EEVPD) than the standard Gaussian distribution. In Appendix C we show that long tails of the distribution also do not increase the bias. Furthermore, Brownian motion problem has a similar structure (hierarchical Bayesian problem) as German Credit and Funnel problems, both of which have smaller bias than the standard Gaussian.

**Initialization bias diagnostics**  The situation should be compared to the issue of initialization bias in adjusted methods, where there is no way to prove that the bias has vanished, and instead, diagnostics are used to identify the unacceptable bias after the chain is already run. This is commonly done by running chains with different initialization and comparing their variances to calculate the Gelman-Rubin $\hat{R}$ (Gelman & Rubin, 1992).

**Discretization bias diagnostics**  Similarly, we encourage the users to run diagnostics to identify cases where the discretization bias is non-negligible. This can be done by running chains with a different step size. One option is to independently run three chains with an equal number of steps: one with the step size as determined by the tuning scheme of Section 6 and two with half of that step size. Running a few parallel MCMC chains for validation purposes is a common practice (Margossian & Gelman, 2024), even for Metropolis adjusted chains and is therefore not a drastic change to the standard MCMC workflow. The samples from the half step size chains are then combined so that their effective sample size is comparable to that of the full step size chain. One can then compare the expectation values computed with the half step size samples and the full steps size samples. Since the discretization bias typically strongly depends on the step size, this test can identify if the discretization bias affects the expectation values at an unacceptable level.

## 9. Conclusions

We have shown that unadjusted gradient-based samplers can be turned into fully automatic black-box algorithms. With our tuning scheme, unadjusted HMC, LMC and MCLMC do not need manual tuning and consistently outperform adjusted methods like NUTS, especially in high dimensions. Their gains are consistent across Bayesian inference benchmark problems and real world applications to quantum and statistical field theory problems. We do not show any pure machine learning applications in this work as it is it controversial whether or not the Bayesian neural networks offer any advantage over the standard neural networks.

Consistent performance makes unadjusted schemes a scalable alternative for a broad range of applications, from probabilistic programming to computational science, and opens new directions for efficient Bayesian inference.

## Acknowledgements

This material is based upon work supported in part by the Heising-Simons Foundation grant 2021- 3282 and in part by the U.S. Department of Energy, Office of Science, Office of Advanced Scientific Computing Research under Contract No. DE-AC02-05CH11231 at Lawrence Berkeley National Laboratory to enable research for Data-intensive Machine Learning and Analysis.

## Impact Statement

This paper presents work whose goal is to advance the field of Machine Learning. There are many potential societal consequences of our work, none which we feel must be specifically highlighted here.

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

```

## A. Covariance Matrix Error

Let SPD($d$) be the space of symmetric positive definite matrices of size $d \times d$.

**Lemma A.1.** $b_{cov}^2(A, B) = \frac{1}{d} \text{Tr}\{(I - A^{-1}B)^2\}$ *is a divergence on SPD(d), meaning that for all $A, B \in$ SPD(d),*

1. $b_{cov}^2(A, B) \geq 0$

2. $b_{cov}^2(A, B) = 0$ *if and only if $A = B$.*

*Proof.* Without loss of generality, we may assume that $A$ is diagonal with positive entries, because the trace is invariant under the change of basis and $A$ must be diagonal in some basis because it is positive-definite. $R \equiv I - A^{-1}B$ then has $R_{ii} = 1 - A_{ii}^{-1}B_{ii}$ on the diagonal and $R_{ij} = -A_{ii}^{-1}B_{ij}$ off the diagonal. The trace is

$$\text{Tr}\{R^2\} = \sum_{i=1}^{d}(1 - A_{ii}^{-1}B_{ii})^2 + \sum_{i \neq j} A_{ii}^{-1}B_{ij}A_{jj}^{-1}B_{ji},$$

where the first and the second term are the contribution from the diagonal and off-diagonal elements respectively. Both terms are non-negative: the first term because it is a sum of squares, the second because all factors $A_{ii}$, $A_{jj}$ and $B_{ij}B_{ji} = (B_{ij})^2$ are non-negative. This already proves (1).

The implication $b_{cov}^2(A, A) = 0$ in (2) is trivial. To prove the other implication in (2), now suppose $b_{cov}^2(A, B) = 0$. This implies that both terms in the above equation are zero, as they are both non-negative. In the second term, $A_{ii} > 0$ are non-zero for any $i$, so the term can only be zero if $B_{ij} = 0$ for any $i \neq j$. The first term can only be non-zero if $A_{ii} = B_{ii}$ for any $i$. We have shown that $A_{ij} = B_{ij}$ for any $i, j$, thus $A = B$. $\square$

Note that the above result is not obvious from the fact that we are computing a trace of the matrix squared. There are non-zero matrices whose square is zero.

$b_{cov}^2$ has another nice property – it can be related to the effective sample size in a covariance matrix independent way:

**Lemma A.2.** *Let $\boldsymbol{x}^{(k)}$ for $k = 1, 2 \ldots n$ be exact i.i.d samples from $p = \mathcal{N}(0, \Sigma)$ and let $\bar{\Sigma} = \frac{1}{n}\sum_{k=1}^{n} \boldsymbol{x}^{(k)}(\boldsymbol{x}^{(k)})^T$, be the empirical estimate for $\Sigma$. Then*

$$\mathbb{E}_{MC}[b_{cov}^2(\Sigma, \bar{\Sigma})] = (d+1)/n,$$

*where $\mathbb{E}_{MC}[\cdot]$ is the expectation with respect to the sample realizations.*

*Proof.* The empirical estimate is unbiased:

$$\mathbb{E}_{MC}[\bar{\Sigma}] = \frac{1}{n}\sum_{k=1}^{n} \mathbb{E}_{MC}[\boldsymbol{x}^{(k)}(\boldsymbol{x}^{(k)})^T] = \Sigma,$$

and

$$\mathbb{E}_{MC}[\bar{\Sigma}_{ab}\bar{\Sigma}_{cd}] = \Sigma_{ab}\Sigma_{cd} + \frac{1}{n}\big(\Sigma_{ac}\Sigma_{bd} + \Sigma_{ad}\Sigma_{bc}\big).$$

The expected error of the empirical covariance matrix is then:

$$\mathbb{E}_{MC}[b_{cov}^2] = 1 - 2\frac{1}{d}[\Sigma^{-1}]_{ij}\mathbb{E}_{MC}[\bar{\Sigma}_{ij}] + \frac{1}{d}[\Sigma^{-1}]_{ij}\mathbb{E}_{MC}[\bar{\Sigma}_{jk}\bar{\Sigma}_{li}][\Sigma^{-1}]_{kl}$$

$$= 1 - 2 + \frac{1}{d}[\Sigma^{-1}]_{ij}\bigg(\Sigma_{jk}\Sigma_{li} + \frac{1}{n}\big(\Sigma_{jl}\Sigma_{ki} + \Sigma_{ji}\Sigma_{kl}\big)\bigg)[\Sigma^{-1}]_{kl} = \frac{d+1}{n},$$

where Einstein's convention of summing over the repeated indices was used. $\square$

This result implies that for Gaussian distributions, $b_{cov}^2$ could be used to define the effective sample size (ESS) of the estimate. Concretely, given some samples whose empirical covariance matrix error is $b_{cov}^2$, we can define their effective sample size to be $n_{\text{eff}} = (d+1)/b_{cov}^2$, i.e., this is the number of exact samples that would yield the same covariance matrix error as given samples. For example, given a target distribution $p$ in $d = 9$, suppose it takes 1000 samples from a Markov

chain to achieve $b^2_{cov} = 0.1$. This would correspond to $(9+1)/0.1 = 100$ effective samples, so $0.1$ effective samples per step. Although this result rigorously only holds for Gaussian target distributions it offers some interpreteability to the value of $b_{cov}$.

Note that we could also take the more transparently non-negative definition $b^2_F(A, B) = \frac{1}{d}\operatorname{Tr}\{(I - A^{-1}B)((I - A^{-1}B)^T\} = \frac{1}{d}\|I - A^{-1}B\|^2_F$, where $\|\cdot\|_F$ is the Frobenius norm. However, $b^2_F$ cannot be related to the effective sample size in a covariance matrix independent way. Instead,

$$\mathbb{E}_{MC}[b^2_F] = \frac{1}{d}\mathbb{E}_{MC}[\big(\delta_{ij} - [\Sigma^{-1}]_{ik}\bar{\Sigma}_{kj}\big)\big(\delta_{ij} - [\Sigma^{-1}]_{il}\bar{\Sigma}_{lj})] \tag{12}$$

$$= 1 - 2\frac{1}{d}[\Sigma^{-1}]_{ij}\mathbb{E}_{MC}[\bar{\Sigma}_{ij}] + \frac{1}{d}[(\Sigma^{-1})^2]_{ij}\mathbb{E}_{MC}[\bar{\Sigma}^2]_{ji} = \frac{1}{n}\Big(1 + \frac{1}{d}\operatorname{Tr}\{\Sigma\}\operatorname{Tr}\{\Sigma^{-1}\}\Big),$$

so we will note use this definition here.

Finally, $b_{cov}$ is invariant to the linear change of basis

**Lemma A.3.** *For an invertible matrix $A$, and a change of basis $x' = Ax$, the covariance matrix error does not change, that is, $b^2_{cov}(\Sigma_{p(x)}, \Sigma_{q(x)}) = b^2_{cov}(\Sigma_{p(x')}, \Sigma_{q(x')})$.*

*Proof.* The covariance matrix transforms as

$$[\Sigma_{p(x')}]_{ij} = \int x'_i x'_j p(x') dx' = [A\,\Sigma_{p(x)}A^T]_{ij},$$

hence the covariance matrix bias is invariant:

$$b^2_{cov}(\Sigma_{p(x)}, \Sigma_{q(x)}) = \frac{1}{d}\operatorname{Tr}\Big\{\big(I - (A\Sigma_{p(x)}A^T)^{-1}(A\Sigma_{q(x)}A^T)\big)^2\Big\}$$

$$= \frac{1}{d}\operatorname{Tr}\Big\{\big(I - (A^T)^{-1}\Sigma^{-1}_{p(x)}A^{-1})(A\Sigma_{q(x)}A^T)\big)^2\Big\}$$

$$= \frac{1}{d}\operatorname{Tr}\Big\{\big(I - (A^T)^{-1}\Sigma^{-1}_{p(x)}\Sigma_{q(x)}A^T\big)^2\Big\}$$

$$= \frac{1}{d}\operatorname{Tr}\Big\{(A^T)^{-1}\big(I - \Sigma^{-1}_{p(x)}\Sigma_{q(x)}\big)^2 A^T\Big\}$$

$$= b^2_{cov}(\Sigma_{p(x')}, \Sigma_{q(x')}).$$

$\square$

# B. Proofs

## B.1. Lemma 4.1: EEVPD

*Proof.* We will work in the eigenbasis, where the dynamics is decoupled. It then suffices to analyze each dimension separately. Let $x_i(t)$ and $u_i(t)$ be components of $x(t)$ and $u(t)$ along the dimension that we analyze and $\sigma^2_i$ the eigenvalue of the covariance matrix in that direction. The velocity Verlet integrator update (4) with step size $\epsilon$ can be written compactly as (Gouraud et al., 2025)

$$\Phi_\epsilon(z_i) = \begin{bmatrix} \cos h & \alpha\sigma_i \sin h \\ -\alpha^{-1}\sigma_i^{-1}\sin h & \cos h \end{bmatrix} z_i \equiv Az_i.$$

Here, $z_i(t) = (x_i(t), u_i(t))$, $\alpha = (1 - y_i/4)^{-1/2}$, $y_i = \epsilon^2/\sigma^2_i$ and $\sin h = \sqrt{y_i}/\alpha$. The energy error is

$$\Delta^i_H \equiv \Delta_H(\Phi_\epsilon(z_i), z_i) = \frac{1}{2}\Phi_\epsilon(z_i)^T D\Phi_\epsilon(z_i) - \frac{1}{2}z_i^T Dz_i = \frac{1}{2}z_i^T Mz_i,$$

where $D = \operatorname{Diag}(1/\sigma^2_i, 1)$ and

$$M = A^T DA - D = (1 - \alpha^2)\begin{bmatrix} (\alpha\sigma_i)^{-2}\sin^2 h & -(\alpha\sigma_i)^{-1}\sin h \cos h \\ -(\alpha\sigma_i)^{-1}\sin h \cos h & -\sin^2 h \end{bmatrix}.$$

Denote by $\tilde{p}_i$ the stationary distribution for $\mathbf{z}_i$, namely $x_i \sim \mathcal{N}(0, \tilde{\sigma}_i)$, $u_i \sim \mathcal{N}(0,1)$, where $\tilde{\sigma}_i = \sigma_i/\alpha$ is taken from Equation (10).

The contribution to the EEVPD from $\mathbf{z}_i$ is

$$\mathrm{Var}_{\tilde{p}_i}[\Delta_H^i] = \mathbb{E}_{\tilde{p}_i}[(\Delta_H^i)^2] - \mathbb{E}_{\tilde{p}}[\Delta_H^i]^2.$$

The expectation value in the second term is

$$\mathbb{E}_{\tilde{p}}[\Delta_H^i] = \frac{1}{2}\big(M_{11}\mathbb{E}_{\tilde{p}}[x^2] + M_{22}\mathbb{E}_{\tilde{p}}[u^2]\big)$$

and for the first

$$\mathbb{E}_{\tilde{p}}[(\Delta_H^i)^2] = \frac{1}{4}\mathbb{E}_{\tilde{p}}[(M_{11}x^2 + 2M_{12}xu + M_{22}u^2)^2]$$

$$= \frac{1}{4}\big(M_{11}^2\mathbb{E}_{\tilde{p}}[x^4] + M_{22}^2\mathbb{E}_{\tilde{p}}[u^4] + (4M_{12}^2 + 2M_{11}M_{22})\mathbb{E}_{\tilde{p}}[x^2]\mathbb{E}_{\tilde{p}}[u^2]\big).$$

In both expressions we have dropped the vanishing contributions that contain $\mathbb{E}_{\tilde{p}}[xu] = 0$, $\mathbb{E}_{\tilde{p}}[xu^3] = 0$ or $\mathbb{E}_{\tilde{p}}[x^3u] = 0$. Combining both terms together and using $\mathbb{E}_{\tilde{p}}[x^4] = 3\mathbb{E}_{\tilde{p}}[x^2]^2$ and $\mathbb{E}_{\tilde{p}}[u^4] = 3\mathbb{E}_{\tilde{p}}[u^2]^2$ we get

$$\mathrm{Var}[\Delta_H^i] = \frac{1}{4}\big(2M_{11}^2\mathbb{E}_{\tilde{p}}[x^2]^2 + 2M_{22}^2\mathbb{E}_{\tilde{p}}[u^2]^2 + 4M_{12}^2\mathbb{E}_{\tilde{p}}[x^2u^2]\big).$$

Inserting $\mathbb{E}_{\tilde{p}}[x^2] = \sigma^2\alpha^2$ and $\mathbb{E}_{\tilde{p}}[u^2] = 1$ gives

$$\mathrm{Var}[\Delta_H^i] = \frac{(1-\alpha^2)^2}{2}\big(\sin^4 h + \sin^4 h + 2\sin^2 h\cos^2 h\big) = (1-\alpha^2)^2\sin^2 h = \frac{y^3}{16(1-y/4)},$$

which is $E(y)$ from the statement of the theorem. EEVPD is thus

$$\mathrm{EEVPD} = \frac{1}{d}\sum_{i=1}^d \mathrm{Var}[\Delta_H^i] = \frac{1}{d}\sum_{i=1}^d E(y_i).$$

$\square$

### B.2. Theorem 4.2: bias bounds

*Proof.* Let's start with the covariance matrix bias. Due to Equation (10), the asymptotic covariance error (6) is

$$b_{cov}^2(\Sigma, \tilde{\Sigma}) = \frac{1}{d}\sum_{i=1}^d \big(1 - (1 - \epsilon^2/4\sigma_i^2)^{-1}\big)^2 = \frac{1}{d}\sum_{i=1}^d B(\epsilon^2/\sigma_i^2),$$

where $B(y) = \frac{y^2}{16(1-y/4)^2}$. We thus see that $\varphi$ from the statement of the theorem is $\varphi = E \circ B^{-1}$. Lemma B.1 shows that $\varphi(x)$ restricted to $0 < x < 11 - 4\sqrt{7}$ is a convex, monotonically increasing function. By Jensen's inequality this implies that

$$\varphi(b_{cov}(\Sigma, \tilde{\Sigma})^2) \le \mathrm{EEVPD},$$

as long as $\varphi(b_{cov}^2) < \varphi(11 - 4\sqrt{7}) = (-134 + 52\sqrt{7})/9 \approx 0.397674$. The assumption of the theorem that EEVPD < 0.397 is a sufficient condition for this to hold. Since $\varphi(x)$ is not a linear function, Jensen's inequality becomes an equality if and only if $\sigma_i = \sigma_j$ for all $i, j$. $\varphi$ is a monotonically increasing function so it is a bijection and its inverse is also a monotonically increasing function. The inverse can therefore be applied to both sides of the above inequality to yield the desired result.

The proof of the Wasserstein distance part of the theorem is similar. For zero-mean Gaussian distributions with covariance matrices $\Sigma_p$ and $\Sigma_q$, the Wasserstein distance reduces to (Olkin & Pukelsheim, 1982)

$$\mathcal{W}_2(\mathcal{N}(0, \Sigma_p), \mathcal{N}(0, \Sigma_q))^2 = \mathrm{Tr}\Big\{\Sigma_p + \Sigma_q - 2\big(\Sigma_p^{1/2}\Sigma_q\Sigma_p^{1/2}\big)^{1/2}\Big\}.$$

By using Equation (10) for unadjusted HMC or LMC with the velocity Verlet integrator we therefore get

$$\mathcal{W}_2(p, \tilde{p})^2 = \epsilon^2 \sum_{i=1}^{d} W(\epsilon^2/\sigma_i^2),$$

where

$$W(y) = \frac{2(1 - y/8 - \sqrt{1 - y/4})}{y(1 - y/4)}$$

is as in the statement of the theorem. Lemma B.2 shows that $\varphi_W(x)$ is a convex, monotonically increasing function for $0 < \varphi_W(x) < 27/4 = 6.75$. By Jensen's inequality this implies that

$$\varphi_W(w) \leq \text{EEVPD},$$

as long as EEVPD $< 6.75$. Here $w = \mathcal{W}_2(p, \tilde{p})^2/d\epsilon^2$. Since $\varphi_W$ is not a linear function, Jensen's inequality becomes an equality if and only if $\sigma_i = \sigma_j$ for all $i, j$. $\varphi_W$ is a monotonically increasing function so it is a bijection and its inverse is also a monotonically increasing function. The inverse can therefore be applied to both sides of the above inequality to yield

$$w \leq \varphi_W^{-1}(\text{EEVPD}).$$

$\square$

## B.3. Convexity

**Lemma B.1.** $\varphi(x) = 4x^{3/2}/(1 + x^{1/2})^2$ is monotonically increasing for $x > 0$ and convex for $0 < x < 11 - 4\sqrt{7}$.

*Proof.* $\varphi(x)$ is monotonically increasing because its derivative

$$\varphi'(x) = \frac{2x^{1/2}(3 + x^{1/2})}{(1 + x^{1/2})^3}$$

is positive (all terms are positive). To show that it is convex, we compute its second derivative

$$\varphi''(x) = \frac{3 - 4x^{1/2} - x}{x^{1/2}(1 + x^{1/2})^4}$$

The denominator is positive for $x > 0$. The numerator is a quadratic polynomial $p(y) = 3 - 4 - y^2$ in $y = \sqrt{x}$. Its roots are $y_{1,2} = -2 \pm \sqrt{7}$. Since $p(0) = 3 > 0$ the numerator is positive for $0 < y < -2 + \sqrt{7}$ corresponding to $0 < x < (-2 + \sqrt{7})^2 = 11 - 4\sqrt{7}$ so $\varphi(x)$ restricted to this interval is convex. $\square$

**Lemma B.2.** $\varphi_W(x)$ from Theorem 4.2 is monotonically increasing and convex for $0 < \varphi_W(x) < 27/4$.

*Proof.* To prove that $\varphi_W$ is monotonically increasing we will show that its derivative is positive. We cannot solve for for $W^{-1}$ explicitly, but nontheless

$$\varphi'_W(w) = E'(W^{-1}(w))(W^{-1})'(w) = \frac{E'(y)}{W'(y)},$$

where we have denoted $y = W^{-1}(w)$. We will show that both denominator and numerator are positive for $0 \leq y < 4$, i.e. in the range where the velocity Verlet integrator is stable.

The numerator is

$$E'(y) = \frac{3y^2(1 - y/6)}{(1 - y/4)^2}$$

which is positive for $0 < y < 4$.

To simplify the denominator we use the reparametrization $y = 4\sin^2 \xi$, such that $0 \leq \xi < \pi/2$. In the new parametrization

$$W(\xi(y)) = \frac{\sin^4(\xi/2)}{4\sin^2(2\xi)}.$$

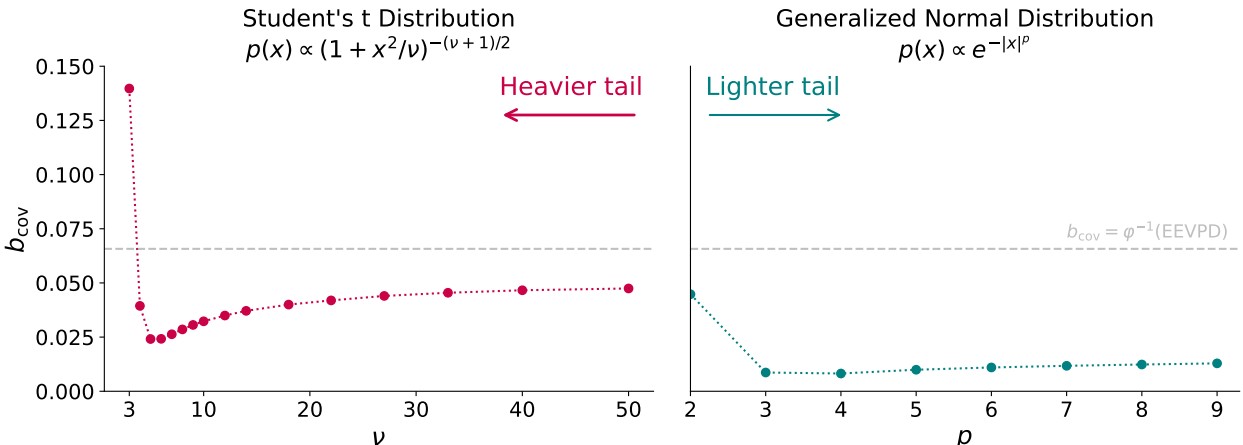

*Figure 7.* Asymptotic bias as a function of the weight of the tails of the distribution. The bias bound from Theorem 4.2, which is valid for Gaussians with unadjusted HMC is shown for reference. Standard Gaussian, which is the limit of $\nu \to \infty$ and $p \to 2$ is a local maximum of the bias.

We get

$$W'(y) = \frac{1 + \cos\xi - \cos^2\xi}{128\cos^4(\xi/2)\cos^4(\xi)} \geq \frac{\cos\xi}{128\cos^4(\xi/2)\cos^4(\xi)} > 0,$$

where we have used that $-\cos^2\xi > -1$.

To prove that $\varphi_W$ is convex, we will show that its second derivative is positive. We have

$$\varphi_W''(w) = E''(W^{-1}(w))((W^{-1})'(w))^2 + E'(W^{-1}(w))(W^{-1})''(w) = \frac{E''(y)}{W'(y)^2} - \frac{E'(y)W''(y)}{W'(y)^3}$$

Which after some algebra reduces to

$$\varphi_W''(y) = \frac{(2-s)(s^2-1)(s^4+4s^3+7s^2+6s-6)}{8(s^2+s-1)},$$

where we have first reparametrized $y(\xi) = 4\sin^2\xi$ and then $s(\xi) = 1/\cos(\xi)$. The range $0 < y < 4$ corresponds to $0 < \xi < \pi/2$ which in turn corresponds to $1 < s < \infty$.

All the factors except for $(2-s)$ are non-negative at $s = 1$ and monotonically increasing for $s > 1$, so they are all positive for $s > 1$. The second derivative is then positive for $1 < s < 2$, corresponding to $0 < \xi < \pi/3$ or $0 < y < 3$. $\varphi_W(x)$ is thus monotonically increasing and convex for $0 < \varphi_W(x) < E(3) = 27/4$. $\square$

## C. Bound Sensitivity

We here study the sensitivity of the bias bound to the tails of the distribution. We study two one parametric families of distributions that reduce to the standard Gaussian in one limit:

- Student's t distribution with a parameter $\nu$ has a probability density $p(x) \propto (1 + x^2/\nu)^{-(\nu+1)/2}$. In the limit $\nu \to \infty$ this reduces to the standard Gaussian. With decreasing $\nu$ the tails of the distribution get heavier. In the $\nu = 2$ extreme we obtain the Cauchy distribution whose tails are so heavy that the second moments no longer exist.

- Generalized Normal distribution with a parameter $p$ has a probability density $p(x) \propto e^{-|x|^p}$. In the limit $p = 2$ this reduces to the standard Gaussian. With increasing $p$ the tails of the distribution get lighter.

We take the 100 dimensional products of these distributions to study targets in 100 dimensions. On a grid of parameters $\nu$ and $p$ we then run unadjusted MCLMC and fix the step size so that EEVPD equals $10^{-3}$. We then run very long chains with

this step size (50 million steps) and check that the covariance matrix bias no longer decays. Figure 7 shows the resulting asymptotic bias as a function of $\nu$ and $p$. It can be seen that the standard Gaussian is a local maximum of the bias at a fixed EEVPD, i.e., either increasing or decreasing the weight of the tails reduces the bias. The bias does start to increase again as we approach $\nu = 2$, which could be due to the proximity of the failure of the bias definition, as the second moments no longer exist for $\nu = 2$.

## D. Bias-Variance Tradeoff

We have shown how to control the covariance matrix bias to be below some desired threshold. However, typically bias is not of direct interest, but instead we want to control the error of the expectation values, which additionally contains the variance, i.e. the second term in Equation (2). The variance of a stationary chain is

$$\text{Var}[\bar{f}] = \text{Var}_p[f](1 + 2\sum_{k=1}^{n}(1 - k/n)\rho_k)/n \equiv \text{Var}_p[f]\tau_{\text{int}}/n, \tag{13}$$

where the autocorrelation coefficients are $\rho_k = \mathbb{E}[(f(x_i) - \mathbb{E}_p[f])(f(x_{i+k}) - \mathbb{E}_p[f])]/\text{Var}_p[f]$.

We would therefore like to know how to optimally set the bias, given some error tolerance. Here we will provide a heuristic, based on estimating the second moment $\mathbb{E}[x^2]$ of a one-dimensional standard Gaussian target with velocity Verlet unadjusted HMC.

In this case, $\rho_k = \rho^k$ (Gouraud et al., 2025), where

$$\rho = \cos^2\left(\frac{T}{\epsilon}\arcsin(\alpha\epsilon/\sigma)\right). \tag{14}$$

This makes the sum in Equation (13) expressable in terms of geometric series:

$$\tau_{\text{int}} = 1 + 2S(\rho) - 2\rho S'(\rho)/n = \frac{1 + \rho}{1 - \rho}\left(1 - \frac{2\rho}{n}\frac{1 - \rho^n}{1 - \rho^2}\right), \tag{15}$$

where $S(\rho) = \sum_{k=1}^{n}\rho^k = \rho(1 - \rho^n)/(1 - \rho)$. The variance at the lowest order in the small-$\epsilon$-expansion is then

$$\text{Var}[x^2] \asymp \frac{2L}{N\epsilon}\lim_{\epsilon \to 0}\tau_{\text{int}}(\epsilon) = c_v/\epsilon, \tag{16}$$

where $c_v$ is constant, independent of the step size and $\asymp$ denotes asymptotic equivalence as $\epsilon \to 0$. The bias of the second moment in this limit is

$$\text{Bias}[x^2] = \frac{1}{1 - \epsilon^2/4} - 1 \asymp \epsilon^2/4 = c_b\epsilon^2, \tag{17}$$

where we have used Equation (10) and denoted $c_b = 1/4$.

Combining Equations (16) and (17) we get for the RMSE in the small step size limit:

$$\text{RMSE}^2 = c_b\epsilon^4 + c_v/\epsilon, \tag{18}$$

where $c_b$ and $c_v$ are defined above and are independent of the step size. We would like to set the step size so that it minimizes the RMSE. The optimum is found at

$$0 = \frac{d}{d\epsilon}\text{RMSE}^2 = 4c_b\epsilon^3 - c_v/\epsilon^2, \tag{19}$$

which gives the optimal step size $\epsilon_{\text{opt}} = (c_v/4c_b)^{1/5}$. At the optimal step size, bias squared is one fifth of the error squared:

$$\frac{\text{Bias}^2}{\text{RMSE}^2} = \frac{1}{1 + \frac{c_v/\epsilon_{\text{opt}}}{c_v\epsilon_{\text{opt}}^4}} = \frac{1}{5}. \tag{20}$$

This is the prescription that we use in Table 2 and in the numerical experiments of Section 7. It is based on the Gaussian assumption, so it might suboptimal for the non-Gaussian distributions. However, the samples would still eventually converge below the required RMSE, as long as the EEVPD-based bound holds. We note that the results in Section 7 do not even exhibit a significant decrease in efficiency compared to the grid search results and all unadjusted samplers converge to the desired accuracy.

| relative RMSE tolerance | Bias tolerance | EEVPD |
|:---:|:---:|:---:|
| 50% | 22% | $3.0 \times 10^{-2}$ |
| 10% | 4.5% | $3.3 \times 10^{-4}$ |
| 5% | 2.2% | $4.3 \times 10^{-5}$ |
| 1% | 0.45% | $3.5 \times 10^{-7}$ |

*Table 2.* Tabulated values of EEVPD (third column) that ensure a desired asymptotic covariance matrix bias (second column). A useful, quick recipe to compute approximation for small $b_{cov}^2$ is EEVPD $= \varphi(b_{cov}^2) \approx 4b_{cov}^3$. Optimal asymptotic bias should be smaller than the given relative root mean square error tolerance (Equation (1)), for example, one can use the prescription $\text{Bias}^2 < \text{RMSE}^2/5$ from Appendix D. This error is given in the first column.

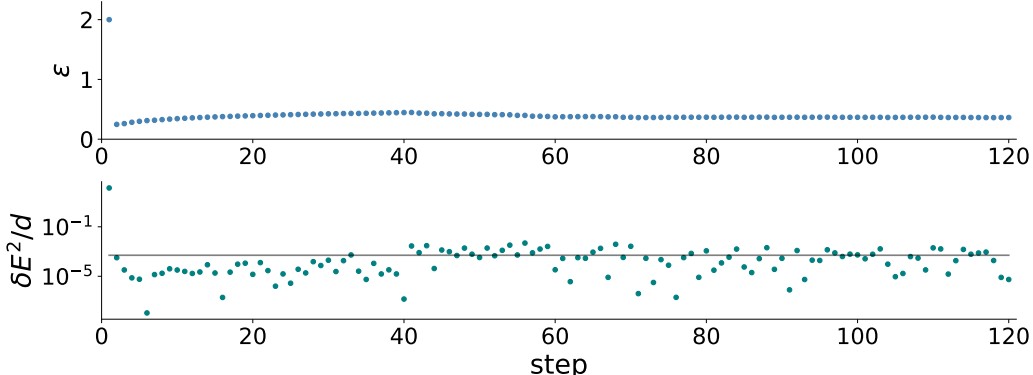

*Figure 8.* The step size tuning algorithm from Section E, applied to the Rosenbrock target distribution in $d = 36$ with MCLMC sampler. The sequential algorithm was initialized from the standard Gaussian distribution with a random initial velocity orienation. Top: the step size as a function of leapfrog integration steps. Bottom: per dimension squared energy error for each step. The algorithm quickly converges to the targeted EEVPD = 0.001, shown with a black line.

## E. Step Size Tuning

We here describe the scheme for quickly tuning the step size $\epsilon$ to achieve a desired EEVPD. We verified that it indeed yields the desired EEVPD in the experiments performed in this work, but we do not provide any convergence guarantees. More generally, one can instead use other stochastic optimization routines, for example the dual averaging algorithm (Hoffman & Gelman, 2014; Nesterov, 2009).

Suppose we did a leapfrog step with size $\epsilon_k$ and found some energy error $\Delta_H^k$. Using only this knowledge and the scaling from Equation (4.1) for small step sizes, $\Delta_H \propto \epsilon^6$, we could estimate the optimal step size to use in the next iteration as $\epsilon_{k+1} = \xi_k^{-1/6}$ where

$$\xi_k = \frac{(\Delta_H^k)^2}{d} \frac{1}{\alpha \, \epsilon_k^6} \tag{21}$$

and $\alpha$ is the desired EEVPD. As we do more leapfrog steps, we can improve our estimate by averaging the energy errors.

---

**Algorithm 1** Step size tuning

---

**Input:** initial condition $(\boldsymbol{x}, \boldsymbol{u})$; initial step size $\epsilon > 0$; number of integration steps $N > 0$; desired EEVPD $\alpha > 0$.
**Output:** step size $\epsilon$.
$A \leftarrow 0, \ B \leftarrow 0$
**for** $n \leftarrow 0$ **to** $N$ **do**
  $(\boldsymbol{x}, \boldsymbol{u}), \Delta E \leftarrow \Phi_\epsilon(\boldsymbol{x}, \boldsymbol{u})$
  $\xi \leftarrow \text{Equation (21)}(\Delta E, \epsilon, \alpha)$
  $A \leftarrow A\gamma + \xi \, w(\xi)$
  $B \leftarrow B\gamma + w(\xi)$
  $\epsilon \leftarrow (A/B)^{-1/6}$
**end for**

---

Our estimate of the optimal step size is then a weighted sum:

$$\epsilon_{n+1} = \left( \frac{\sum_{k=1}^{n} w(\xi_k) \gamma^{n-k} \xi_k}{\sum_{k=1}^{n} w(\xi_k) \gamma^{n-k}} \right)^{-1/6}. \tag{22}$$

We have introduced two types of weights:

- The weights $w$ parametrize our trust in the predictions from the too large and too small $\epsilon$. We take the log-normal penalty

$$w(\xi) = \exp\left\{ -\frac{1}{2} (\log \xi)^2 / \sigma_\xi^2 \right\}, \tag{23}$$

  with $\sigma_\xi = 1.5$.

- $\gamma$ is the forgetting factor. It is related to the effective sample size $n$ of the estimate (if $w$ were constant) by $\gamma = \frac{n-1}{n+1}$. $n$ is also the number of steps after which the weights have decayed to $e^{-2} = 0.13$. In general, we don't want $n$ to be too small, so that EEVPD is well determined and yet not too large during the burn-in such that the initially heavily biased estimates are forgotten quickly. We find $n = 50$ to work well on all the experiments considered in this work. An example run is shown in Figure 8.

The pseudocode for the proposed algorithm is shown in 1.

## F. Model Details

### F.1. Benchmarks

The following benchmarks are used:

- A Standard Gaussian in $d = 100$.

- An ill-conditioned Gaussian in $d = 100$ and condition number $\kappa = 1000$. The eigenvalues of the covariance matrix are equally spaced in log.

- A Rosenbrock function with $Q = 0.1$ from (Grumitt et al., 2022). This is a banana shaped target in two dimensions, see Figure 8 in (Robnik et al., 2024). We use a product of 18 independent copies, so the total dimension of the target is 36. An exception is Figure 2 where we study the performance as a function of the number of copies.

- A Funnel problem in 101 dimensions: this is a hierarchical Bayesian model with a funnel shape (Grumitt et al., 2022). The goal is to infer the hierarchical parameter $\theta$ and the latent variables $\{z_i\}_{i=1}^{100}$, given the noisy observations $y_i \sim \mathcal{N}(z_i, 1)$. The prior is Neal's funnel (Neal, 2011): $\theta \sim \mathcal{N}(0, 3)$, $z_i \sim \mathcal{N}(0, e^{\theta/2})$. We set $\theta_{\text{true}} = 0$ and generate the data with the generative process described above. Given this data we then sample from the posterior for $\theta$ and $\{z_i\}_{i=1}^{100}$.

- A Brownian motion example from the Inference Gym (Sountsov et al., 2020), where it is named `BrownianMotionUnknownScalesMissingMiddleObservations`. This is a 32 dimensional hierarchical Bayesian model where Brownian motion with unknown innovation noise and measurement noise is fitted to the noisy and partially missing data.

- The German Credit model, also known as Sparse logistic regression (`GermanCreditNumericSparseLogisticRegression`) is a 51-dimensional Bayesian hierarchical model, where logistic regression is used to model the approval of the credit based on the information about the applicant.

- An Item Response theory model (`SyntheticItemResponseTheory`), which is a 501-dimensional hierarchical problem where students' ability is inferred, given the test results.

- Stochastic Volatility is a 2429-dimensional hierarchical non-Gaussian random walk fit to the S&P500 returns data, adapted from NumPyro (Phan et al., 2019).

The ground truth covariance matrix for the first two problems is known exactly. For the Rosenbrock function, we compute it by drawing exact samples from the posterior. For the other problems, we obtain the ground truth by running very long NUTS chains.

We note that even though some of these benchmark problems are the same as in (Hoffman & Sountsov, 2022) and the evaluation metric is the same, the results cannot be compared, because Hoffman & Sountsov (2022) run a large number of chains in parallel (4096 chains) and combines the samples at each fixed time to compute the expectation values. We on the other hand are interested in the more standard regime where one chain is sequentially run for a longer time and samples at different times are combined to compute the expectation values. This of course results in longer time to convergence, but lower total calculation cost. Furthermore NUTS in (Hoffman & Sountsov, 2022) was adapted to the many-short-chains regime and is therefore not the same algorithm as a more standard NUTS used here.

### F.2. Lattice $\phi^4$ field theory

This is an interacting lattice field theory on the plane. We will adopt its treatment from Robnik & Seljak (2024). In a continuum, the $\phi$ scalar function $\phi(x, y)$ on the plane. The probability density on the field configuration space is proportional to $e^{-S[\phi]}$, where the action is

$$S[\phi(x, y)] = \int \left( - \phi \, \partial^2 \phi + m^2 \phi^2 + \lambda \phi^4 \right) dx dy. \tag{24}$$

The parameters of the theory are the mass $m^2 < 0$, and the quartic coupling $\lambda > 0$.

The system is interesting as it exhibits spontaneous symmetry breaking, and belongs to the same universality class as the Ising model. It does not admit analytic solutions due to the quartic interaction term. It is numerically solved by discretizing the field on a lattice and making the lattice spacing as fine as possible (Gattringer & Lang, 2009). The discretized field is specified by a vector of values on a lattice $\phi_{ij}$ for $i, j = 1, 2, \ldots L$. The dimensionality of the configuration space is $d = L^2$. We will impose periodic boundary conditions, such that $\phi_{i,L+1} = \phi_{i1}$ and $\phi_{L+1,j} = \phi_{1j}$. The lattice action is (Vierhaus, 2010)

$$S_{\text{lat}}[\phi] = \sum_{i,j=1}^{L} 2\phi_{ij}\left(2\phi_{ij} - \phi_{i+1,j} - \phi_{i,j+1}\right) + m^2 \phi_{ij}^2 + \lambda \phi_{ij}^4. \tag{25}$$

We will fix $m^2 = -4$ (which removes the diagonal terms $\phi_{ij}^2$ in the action) as is common (Albergo et al., 2021; Gerdes et al., 2022). In all experiments we fix $\bar{\lambda} = 4$, where $\bar{\lambda} = L^{1/\nu}(\lambda - \lambda_C)/\lambda_C$. Here, $\nu = 1$ and $\lambda_C = 4.25$ are taken from (Vierhaus, 2010). This choice ensures that all of the experiments are performed at approximately the same location in the phase diagram in disordered phase, with minimal dependence on the lattice size (Gerdes et al., 2022; Goldenfeld, 2018).

As in the other experiments, we take the average squared bias of the second moments as the convergence metric, $b_{avg}^2 \equiv d^{-1} \sum_{k,l=1}^{L} b^2(|\widetilde{\phi}_{kl}|^2)$. The only difference is that we here consider the second moments of the Fourier transform $\widetilde{\phi}$ instead of the field itself. The Fourier transform is defined as $\widetilde{\phi}_{kl} = L^{-1} \sum_{nm=1}^{L} \phi_{nm} e^{-2\pi i(kn+lm)/L}$. We use only 2 chains for $\phi^4$ at $1024^2$ dimensions, since this is what fits on a GPU.

### F.3. Lattice U(1) gauge theory

This is a quantum field theory of electrodynamics, such that the gauge symmetry is the U(1) group. It serves as a toy model of the quantum chromodynamics. We will adopt its treatment from (Albergo et al., 2021; Kanwar et al., 2020) and study the theory in two Euclidean dimensions on a $L \times L$ lattice with the periodic boundary conditions. The degrees of freedom are the elements of the $U(1)$ group on each lattice link. The group elements are parametrized by the elements of its Lie algebra, such that an angle $\theta \in [-\pi, \pi)$ is assigned to each lattice link. The configuration space is then parametrized by a vector of link angles: $\Theta = \{\theta_{mn}^i | 1 \leq m \leq L, 1 \leq n \leq L, i \in \{t, x\}\}$. $m$ and $n$ indices determine the start of the link and $i$ its direction (time or space direction). The probability density on the field configuration space $p(\Theta)$ is proportional to $e^{-S[\Theta]}$, where the action is

$$S_{\text{lat}}[\Theta] = -\beta \sum_{m,n=1}^{L} \cos \theta_{mn}^P. \tag{26}$$

Here, the plaquette angle is $\theta_{mn}^P = \theta_{mn}^t + \theta_{m+1,n}^x - \theta_{m,n+1}^t - \theta_{mn}^x$ and the inverse temperature $\beta$ is a free parameter of the theory. In this experiment we set it to $\beta = 2$.

|  | aLMC grid search | uLMC EEVPD | aMCLMC acc. rate | uMCLMC EEVPD | uLMC grid search | NUTS acc. rate |
|---|---|---|---|---|---|---|
| Standard Gaussian | 79,841 | **64,254** | 43,389 | **26,032** | 65,604 | 240,456 |
| Rosenbrock | 659,359 | **415,988** | 540866 | **348,048** | 271,825 | 852,135 |
| Brownian | **93,787** | 112,242 | 76,931 | **41,838** | 73,632 | 146,333 |
| German Credit | 462,843 | **380,569** | 462,770 | **249,674** | 306904 | 756,792 |

*Table 3.* Number of gradient calls needed to get $b_{cov}^2$ below 0.01.

| Model | metric | aLMC | uLMC | aMCLMC | uMCLMC | NUTS |
|---|---|---|---|---|---|---|
| **Standard Gaussian** | $b_{avg}$ | 0.42% | 1.02% | 1.51% | 0.77% | 0.13% |
|  | $b_{cov}$ | 0.06% | 0.32% | 0.31% | 0.29% | 0.06% |
| **Rosenbrock** | $b_{avg}$ | 0.07% | 5.69% | 0.06% | 2.05% | 0.02% |
|  | $b_{cov}$ | 0.03% | 1.64% | 0.01% | 0.90% | 0.01% |
| **Brownian Motion** | $b_{avg}$ | 0.37% | 6.14% | 0.38% | 2.77% | 0.16% |
|  | $b_{cov}$ | 0.05% | 4.96% | 0.06% | 0.49% | 0.03% |
| **German Credit** | $b_{avg}$ | 0.06% | 7.49% | 0.10% | 2.25% | 0.04% |
|  | $b_{cov}$ | 0.05% | 11.58% | 0.08% | 0.93% | 0.05% |
| **Item Response** | $b_{avg}$ | 0.62% | 8.08% | 1.23% | 4.11% | 0.44% |
| **Stochastic Volatility** | $b_{avg}$ | 0.04% | 8.88% | 0.06% | 1.87% | 0.02% |

*Table 4.* Relative error associated with Tables 1 and 3.

Only gauge invariant summary statistics make physical sense for a gauge theory, so we will not consider $\mathbb{E}[(\vartheta_{mn}^i)^2]$ for this problem. We instead study closed loops, which are gauge invariant. One natural choice are the Polyakov loops, i.e., closed loops over the Euclidean time direction:

$$P_n(\Theta) = \exp\{i \sum_{m=1}^{L} \vartheta_{m,n}^t\}. \tag{27}$$

Polyakov loops are of direct physical interest because their autocorrelation can be used to infer the static potential $V$ of the quantum field (Gattringer & Lang, 2009):

$$\mathbb{E}_{\Theta \sim p}[P_n(\Theta)P_0(\Theta)] \propto e^{-LaV(an)}. \tag{28}$$

Here $a$ is the lattice spacing in physical units and the expectation value is over the field configurations. We define the average bias as $b_{avg}^2 = L^{-1} \sum_{n=1}^{L} b^2(P_n(\Theta))$.

## G. Further Experiments

### G.1. Covariance matrix bias

Table 3 shows the convergence cost for different samplers with the $b_{cov}$ metric, analogously to Table 1. Here we omit the higher dimensional problems (Item response and Stochastic Volatility), because their covariance matrix is impractically large. Qualitatively the conclusions are as in Section 7:

- Unadjusted versions of the algorithms tend to perform better than their adjusted counterparts.

- EEVPD based tuning yields close to optimal performance when compared to grid search. An exception is the Rosenbrock distribution, where the conservative choice of EEVPD yields some decrease in performance, but the unadjusted algorithm still performs better than the adjusted one.

- MCLMC performs better than LMC.

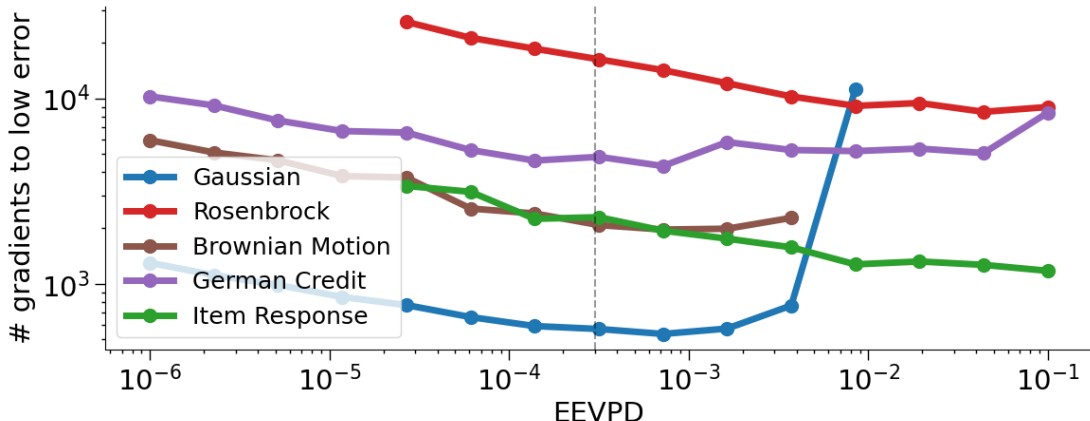

*Figure 9.* Performance of unadjusted LMC as a function of EEVPD (which in turn sets the step size). As can be seen, performance does not change much within a resonable range of EEVPD. The value $\text{EEVPD} = 3 \times 10^{-4}$ that we use in Section 7 is shown as a vertical dashed line.

A notable difference is that all samplers need a larger number of gradient evaluations here, because $b_{cov}$ is a more stringent metric than $b_{avg}$.

### G.2. Ablation study

Here, we investigate how performance of unadjusted LMC varies with the EEVPD value being targeted. Figure 9 shows that our choice of $3 \times 10^{-4}$ is within the safe range for problems that we consider, albeit it is somewhat conservative for the Item Response and Rosenbrock problems.

### G.3. Uncertainity

The errors corresponding to Tables 1 and 3 are shown in Table 4. They are calculated by bootstrap: for a given model, we produce a set of chains (at least 128), and calculate the bias $b_{avg}$ or $b_{cov}$ at each step of the chain. We then resample (with replacement) 100 times from this set, and compute our final metric (number of gradients to low bias) 100 times. We take the standard deviation of this list of length 100 as the estimate of the error and report it relatively to the values in Tables 1 and 3.

## H. Reproducibility

**Code**   All the samplers considered and their tuning schemes are implemented in BlackJax (Cabezas et al., 2024), which is publicly available: `https://blackjax-devs.github.io/blackjax/`. General purpose benchmarking code and experiments in this paper are also publicly available `https://github.com/reubenharry/sampler-benchmarks`.

**Computing architecture**   $\phi^4$ field theory example was run on the NVIDIA A100 GPU (40GB). The other experiments were run on 128 CPU cores, where each core is a 2x AMD EPYC 7763 (Milan) CPU.

