# OpenReview forum: "Practical and Scalable Hamiltonian Monte Carlo Without the Metropolis Test"
_ICML.cc/2026/Conference — ICML 2026 regular_

### Official Review · Reviewer_bTrJ · 2026-02-17

**Soundness:** 2
**Presentation:** 2
**Significance:** 3
**Originality:** 3
**Overall Recommendation:** 5
**Confidence:** 4

**Summary:**

The paper proposes a novel strategy to reduce the bias in Hamiltonian Monte Carlo (HMC) samples, with the goal of avoiding the Metropolis correction step that is typically required when using a finite integration step size. The approach relies on adaptively controlling the step size according to a prescribed tolerance, leveraging the energy change induced by the discretization error of the numerical integrator. The authors derive analytical results in the Gaussian case, establishing a bound that relates the proposed EEVPD quantity to the bias in the estimated covariance matrix. They then provide numerical evidence on a non-Gaussian example suggesting that a similar relationship holds more generally. Motivated by these findings, they introduce an adaptive step-size scheme based on a target EEVPD level, and present numerical experiments for different target values, corresponding to different bias bounds.

**Compliance With Llm Reviewing Policy:**

Affirmed.

**Final Justification:**

My concerns have been adequately addressed, also in responses to other reviews. Hence I raised my score to acceptance.

**Key Questions For Authors:**

1. On the scope of Theorem 4.2:
The Gaussian bound established in Theorem 4.2 is shown to fail in at least one non-Gaussian example (Figure 3). Can the authors precisely characterize the class of target distributions for which the bound is expected to remain informative? Is there a structural property (e.g., log-concavity, quadratic tails, spectral gap conditions) that distinguishes the regimes where the result is meaningful from those where it breaks down?
2. On the representativeness of the counterexample:
Is the non-Gaussian example where the bound fails genuinely pathological, or does it represent a broad and practically relevant class of distributions? Conversely, are the positive non-Gaussian experiments “close to Gaussian” in a quantifiable sense? A systematic analysis of this point would clarify whether the Gaussian result provides a robust design principle or only a heuristic guideline.
3. On extrapolation beyond theory:
Given that the theoretical guarantee holds only in the Gaussian case, what justifies using it to motivate an adaptive step-size strategy in general non-Gaussian settings? Can the authors provide either (i) additional theoretical arguments supporting this extrapolation, or (ii) a reframing of the contribution as a conditional result valid within a well-defined regime?
4. On relation to existing adaptive step-size literature:
The manuscript claims that step-size adaptation is a “notable gap,” yet there exists substantial literature on adaptive step-size selection in MCMC and Hamiltonian Monte Carlo, including both recent machine learning work (e.g., arXiv:2506.04082, arXiv:2410.21587) and earlier statistical physics/lattice field theory contributions (e.g., arXiv:hep-lat/9606020). How does the proposed method differ conceptually and practically from these approaches? Can the authors clarify what is genuinely new relative to existing energy-error- or acceptance-based adaptation strategies?

**Limitations:**

Yes

**Strengths And Weaknesses:**

Soundness:

The paper is sound in its overall objective and in the general methodology adopted to achieve it. However, there are significant concerns regarding the claim that the Gaussian result—particularly the core result in Theorem 4.2—can serve as a robust foundation for scaling the proposed method more broadly.
In Figure 3, the authors already present a counterexample to the hypothesis that the bound established in the Gaussian setting generalizes beyond Gaussian targets. This weakens the logical coherence of the overall narrative. As it stands, the message conveyed to the reader appears somewhat inconsistent: the bound is rigorously established for Gaussian densities and is shown not to hold in at least one non-Gaussian example; nevertheless, it is subsequently used to motivate and design an adaptive step-size strategy for non-Gaussian settings.
A key question that should be addressed is whether the non-Gaussian example where the bound fails is genuinely pathological, or whether instead the cases where the bound appears to hold are “too Gaussian-like” to be representative. Without a clearer characterization of the regimes in which the bound can be expected to remain informative, it is difficult to assess the reliability and scalability of the proposed approach beyond the Gaussian case.
Regarding the experimental part, and the theory, they sound correct per se. What is not convincing is the extrapolation the author do from them.

Presentation:

The presentation would benefit from a substantial reorganization. In its current form, some sections interrupt the logical flow of the manuscript. For instance, the illustrative example in the Introduction appears misplaced and may be better positioned later in the paper, once the theoretical framework has been clearly established.
Moreover, Section 5 seems more closely aligned with the experimental results and could naturally follow Section 6. At present, the manuscript alternates between theoretical developments and numerical illustrations (theory–numerics–theory–numerics), which weakens the narrative coherence. A clearer structural separation between theoretical foundations and experimental validation would likely improve readability and overall impact.

Significance:

The paper addresses a fundamental issue in biased samplers, which is certainly of interest. However, in its present form, the broader impact of the proposed approach is not entirely clear. The main theoretical results appear to hold only for Gaussian densities, a limitation that is also reflected in the numerical experiments. In fact, the presence of counterexamples where the proposed bound is not satisfied suggests that the validity of the result is restricted to a specific class of distributions. This, in itself, is a meaningful and potentially publishable contribution: showing that adaptive time-stepping based on energy differences works under Gaussian assumptions but fails more generally is an important structural insight.
However, the manuscript does not frame the result in this way. Rather than presenting it as a conditional or possibly negative/agnostic result — valid within a well-defined regime — the conclusion suggests that the bound “works almost every time” and may therefore be applicable in a broad context. This extrapolation does not appear to be sufficiently supported by the theoretical analysis, especially in light of the counterexamples. A clearer positioning of the contribution would strengthen the paper. Either the scope should be explicitly restricted to Gaussian (or closely related) settings, or additional theoretical justification should be provided to support claims of broader applicability.

Originality:

To the best of my knowledge, the specific algorithmic formulation proposed in the manuscript may be novel. However, the statement that “a notable gap in the above work is an algorithm for choosing the step size ϵ” is likely overstated in its current form. Adaptive time-step or step-size selection in the context of Markov chain Monte Carlo — particularly Hamiltonian Monte Carlo and related reversible samplers — has a significant literature, including both the machine learning and physics communities.
A more thorough review of adaptive step-size and energy-aware adaptation methods is needed. A preliminary search reveals several recent works that appear directly relevant yet are not cited in the manuscript, for example:
	•	arXiv:2506.04082
	•	arXiv:2410.21587
	•	arXiv:hep-lat/9606020
In addition, there is a long-standing body of work in the physics literature (statistical physics and lattice field theory) addressing step-size adaptation in MCMC and HMC, dating back to at least the 1980s and 1990s. These works investigate how integration error, energy conservation, and acceptance rates interact in practice; they may provide valuable context and points of comparison for the current proposal.
At a minimum, the authors should acknowledge and discuss prior literature on adaptive step-size selection in HMC and related samplers (both modern and classical) and If appropriate, compare empirically with closely related adaptive strategies rather than only against fixed-step baselines.
Without this, the claim that step-size adaptation is a “notable gap” risks overlooking substantial prior work, and the contribution may appear less grounded in the broader research landscape than it actually is.

---

> ### Author Rebuttal · Authors · 2026-03-31
>
> ## Soundness and Significance
> Indeed the provided Gaussian bounds do not always hold in general (even though they often do, as shown in the numerical experiments). Our claim is not that the bounds are guaranteed to work in general, but rather that they often work perfectly in practice (and at least approximately work in all of the problems we considered) making the method useful for the practitioner. Additionally, cases where it fails can be identified by further diagnostics.
>
> This situation is analogous to initialization bias in adjusted methods, where one cannot prove the bias has vanished and must instead rely on diagnostics after running the chain. A common example is the Gelman–Rubin $\widehat{R}$ statistic, which detects dependence on initialization by comparing multiple chains started from different points. If the diagnostic fails, longer chains can be run, though there is no theoretical guidance for choosing the chain length in advance.
> Similarly, for unadjusted methods, we propose using diagnostics to detect non-negligible discretization bias. Specifically, one can run two independent chains with half the step size and compare their combined expectation estimates to those from the original chain. Because discretization bias typically depends strongly on step size, this comparison can reveal whether the bias is unacceptably large. If so, a smaller step size should be used.
>
> We demonstrate that the method works (both in terms of high efficiency and low asymptotic bias) for a range of problems (with and without data, high and low dimensions). The problems where the method works are not “close to Gaussian”, see for example the 2d marginal distribution for the Funnel problem (Figure 4). Rosenbrock function also has long non-Gaussian banana shaped tails (we do not show the plot in the paper, but it can be seen for example in Figure 8 in Robnik et al., Microcanonical Hamiltonian Monte Carlo, 2023). In fact all of these non-Gaussian distributions have lower bias than the standard Gaussian (at a fixed EEVPD). U(1) problem that we have added in response to reviewer 4282 is also highly non-Gaussian (it is not even defined on $\mathbb{R}^d$, but on $S_1^d$). For the discussion of the distribution properties which affect the bias please see our response to the reviewer U12Q.
>
> ## Originality
> There seems to be some confusion that we would like to clarify. Our proposed method is not an “adaptive step size method” in the sense that the step size would be changing during the sampling stage. Instead the step size is adapted only in a tuning stage and is then frozen during the sampling stage. This is analogous to how the step size is tuned in the standard implementations of HMC and NUTS (e.g. in Stan, Numpyro or Blackjax). We have now further clarified this in the text. We have also added the suggested and other relevant references to the adaptive step size algorithms. We have further discussed the literature on step size tuning in connection with the energy error, in particular in molecular dynamics, where unadjusted dynamics is commonly used. There, the energy error is used as a diagnostics: it should not show long term drifts (Tuckerman, Statistical mechanics: theory and molecular simulation, 2023).
>
> Our statement that there is a notable gap in step size tuning literature is in the context of *unadjusted MCMC*, which we now made more explicit. Our work is to our knowledge the only automatic step size tuning algorithm for the unadjusted MCMC. In contrast, Molecular dynamics simulations lack a principled way of determining the stepsize and instead use the knowledge of physics (the relevant time-scales of the bonds) to select the step size and then verify the choice by diagnostics (no energy drift, no divergences). This approach is not suitable for a general purpose use in computational statistics.
>
> ## Presentation
> Thank you for the suggestions which we will gladly implement, in particular we are happy to move the numerics entirely to the end.
>
> ## Summary
> The usefulness of our work is that it brings the unadjusted MCMC sampling to the level of practical applicability of the adjusted MCMC sampling. It does so by providing an automatic step size tuning algorithm for the unadjusted MCMC methods, which has not been achieved before. It is not theoretically guaranteed to work for all distributions, but this is also the case for adjusted MCMC (especially for the finite chain length). Instead we demonstrate numerically that it works. Even in one case where the theoretical bounds for the Gaussians are violated (Brownian motion) the method still achieves the desired low error. The upshot is that the proposed method is consistently orders of magnitudes faster than the adjusted methods in high dimensional settings. We feel that ICML accepts practically minded papers, where a novel method which is demonstrated to perform impressively on challenging benchmarks is regarded as impactful.

---

> > ### Author Rebuttal · Reviewer_bTrJ · 2026-03-31
> >
> > I will raise my score to accept. Thanks for the clarification.

---

### Official Review · Reviewer_U12Q · 2026-03-11

**Soundness:** 3
**Presentation:** 4
**Significance:** 3
**Originality:** 3
**Overall Recommendation:** 5
**Confidence:** 4

**Summary:**

This paper presents an approach to using Hamiltonian Monte Carlo without a Metropolis-Hastings accept-reject step. Such schemes inherently generally have bias (due to lack of the MH acceptance move), but the authors recognize that typically a small amount of bias is acceptable depending on the variance of the estimator itself. To achieve this, the authors develop a quantity (EEVPD) that bounds an appropriate measure of bias and use those results to propose an automatic tuning scheme that achieves a certain asymptotic bias tolerance.

**Compliance With Llm Reviewing Policy:**

Affirmed.

**Final Justification:**

The authors' rebuttal addressed my main concerns. I have decided to increase my score accordingly, as I was primarily wondering about  understanding the main driving factors behind the bias. The paper is in general quite strong and I think would ultimately be useful for practitioners.

**Key Questions For Authors:**

1. For the non-Gaussian distributions in the experiments, we typically see that the Gaussian theory holds approximately for these distributions as well. However, on the Brownian motion example this is not the case. Is there any specific property of this example that could help to inform us as to why this might be the case? In particular, is it possible to take this example further and construct a pathological example for which the Gaussian theory is an arbitrarily bad approximation to the true asymptotic error? I would be interested in understanding the limitations of the current theory and seeing experiments in a "difficult" distribution. Note that, in my opinion, this would only strengthen the paper (by increasing the readers' understanding of where the weaknesses may be).

2. Please address some of the following typos / formatting concerns:
- The abstract seems a bit long. Please try to shorten it in some way.
- Bottom of page 1: "with respect to the Monte Carlo randomness". Please clarify exactly what are the considered sources of randomness (e.g., initialization, Markov transitions, etc.)
- Page 2: "the use of Metropolis-Hastings adjustment ensures that ... the asymptotic bias vanishes, but for finite length chains, it does not remove the variance". There is a strange emphasis here, as it seems to imply that the bias is removed in finite length chains, which is not true. Please rephrase accordingly.
- Bottom of page 2: "Section7" --> "Section 7"
- In equation (5), you are missing a $\circ$ between the composition of the maps.
- In Theorem 4.2, the constants 0.397 and 6.75 seem to appear out of nowhere, with minimal discussion. Please add some comments regarding whether these numbers only appear because of the proof technique, or whether they have actual significance.

3. You mention in the experiments that you "adaptively vary $\epsilon$ to target a desired value of EEVPD". Can you please add more information here unless if I have missed it? How do you properly do this while ensuring that you are not adding further bias to the asymptotic distribution (i.e., in adaptive MCMC you have to be careful about changing the Markov kernel as you proceed)? Otherwise, you have two layers of bias: due to lack of MH acceptance steps, and due to non-measure-preserving adaptive moves.

**Limitations:**

The authors have a separate subsection of the paper dedicated to limitations of the current work. I have no major concerns here, other than my comment above that some additional simulations could be useful to better understand non-Gaussian settings where the theory breaks down.

**Strengths And Weaknesses:**

Overall, the ideas presented in this work seem to be technically sound. For instance, some of the main results such as Theorem 4.2 are insightful, useful, and seem to be technically correct. Overall, I also found the paper quite easy to follow and that content was presented quite clearly.

In my opinion, this paper has its place in the MCMC literature. Often we tend to emphasize the asymptotic variance in MCMC knowing that we achieve an asymptotic bias of zero. However, with uncorrected MCMC schemes, such as the ones studied in this paper, it is logical to consider the tradeoff between asymptotic bias and asymptotic variance. Therefore, I think that this paper points in the right direction and should encourage useful discussion among other MCMC theorists. Perhaps one reason that this has not been considered widely before is because of the technical difficulty of analyzing asymptotic bias for general samplers/distributions. This paper presents some encouraging experimental results on non-Gaussian distributions that could lead to possible conjectures on the asymptotic bias of uncorrected HMC schemes.

I found that this work was quite original and helped to deepen understanding of existing methods.

---

> ### Author Rebuttal · Authors · 2026-03-30
>
> We thank the reviewer for the useful comments and suggestions, which we have incorporated in the text.
>
> ## 1. Driving factors for the bias
> Indeed it is useful to understand which properties of the target distribution reduce the asymptotic bias relative to the energy error and which increase it. We make some observations:
>
> - **Condition number**: theoretical results in the paper show that having different scales for different parameters reduces the bias, i.e. non-standard Gaussian distributions have smaller bias (at fixed EEVPD) than the standard Gaussian distribution.
>
> - **Dimensionality:** Increasing the dimensionality (by means of making multiple copies of the distribution) does not affect the bound by definition.
>
> - **Hierarchical problems:** It is difficult to identify which property of the Brownian motion is responsible for increased bias, since it has a similar structure (hierarchical Bayesian problem) as German Credit and Funnel problems, both of which have reduced bias.
> In light of lack of further understanding we propose to use diagnostics to identify (somewhat rare) cases when the bias is larger than expected from the EEVPD.
>
> The situation is analogous to the problem of initialization bias in adjusted methods. In that setting, there is no way to rigorously prove that the bias has disappeared; instead, one must rely on diagnostics applied after running the chain. For example, the Gelman–Rubin $\widehat{R}$ statistic is commonly used to detect sensitivity to initialization by comparing independent chains.
>
> Similarly, here we propose (currently described in the limitations section, but to be moved to a more prominent place) the use of diagnostics to detect whether discretization bias is significant. Specifically, one can run two independent chains with half the stepsize and compare the expectation values computed from their combined samples with those obtained from the original chain. Because discretization bias typically depends strongly on the stepsize, this comparison can reveal whether it affects the results at an unacceptable level.
>
> ## 2. Presentation
> - **Abstract:** thank you for pointing this out, we have shortened it quite a bit.
>
> - **Page 1:** By Monte Carlo randomness we mean initialization and Markov transition randomness, we have now clarified this in the text.
> - **Page 2:** We have now clarified that, thank you for pointing it out.
> - **Typos:** Thank you, we fixed the typos.
> - **Constants in Theorem 4.2** are actually exact results for the standard Gaussian (up to the number of decimal places provided, exact constants $(−134+52\sqrt{7})/9$  and $27/4$ are provided in the proof). The upper bound together is tight in the sense that it is the equality for the standard Gaussian.
>
> ## 3. Step size adaptation
> We apologize for the confusion, the stepsize is only varied in the adaptation phase and is then frozen during the sampling. This is completely analogous to how the stepsize is tuned in HMC or NUTS. While the adjusted algorithms adapt the stepsize to achieve some desired acceptance probability we adapt the stepsize to achieve a desired EEVPD. In fact, the same general purpose stochastic optimization algorithms can be used to achieve that, for example dual averaging. We develop a specialized algorithm in appendix D.4. which is designed to quickly forget the initial burn-in samples. It is specialized in the sense that it uses the analytical approximation to the scaling of the EEVPD with the stepsize to speed up adaptation.
>
> ## Limitations
> We have now expanded the conclusion to include the discussion of the various properties of the target distribution and how they affect the bias at a fixed EEVPD. Furthermore, as we do not provide a rigorous analysis beyond the Gaussian distribution we advocate the usage of the diagnostics (see above) to identify cases where bias is larger than accepted and note that this is similar to how the initialization bias in adjusted methods is usually treated.

---

> > ### Author Rebuttal · Reviewer_U12Q · 2026-04-02
> >
> > I thank the authors for their responses. My concerns have been addressed and I am raising my score accordingly.

---

### Official Review · Reviewer_4282 · 2026-03-11

**Soundness:** 3
**Presentation:** 3
**Significance:** 2
**Originality:** 2
**Overall Recommendation:** 4
**Confidence:** 4

**Summary:**

This paper proposes an unadjusted version of the HMC algorithm where the step size is chosen automatically, allowing the method to be used as a black-box sampler. The authors study a general area of improving gradient-based samplers by avoiding the Metropolis–Hastings correction. The main idea is to minimise the overall computational cost by allowing a small asymptotic bias introduced by numerical integration while making sure that this bias remains smaller than the Monte Carlo variance. The step size is automatically selected by the method using the EEVPD metric. The study provides theoretical analysis demonstrating that the EEVPD metric (for Gaussian targets) can be used to bound the asymptotic covariance matrix error. Benchmark experiments for HMC, LMC, and MCLMC methods are presented to verify that the covariance matrix bias behaves as predicted by the theory.

**Compliance With Llm Reviewing Policy:**

Affirmed.

**Final Justification:**

My remaining concern is that the theory guarantee is still limited to the Gaussian case, but this now seems like a manageable limitation rather than a decisive weakness. Therefore, I am raising my score to weak accept.

**Key Questions For Authors:**

Could you discuss how this method could be applied when the covariance matrix is not Gaussian and is hard to numerically verify?

**Limitations:**

The experimental evaluation primarily focuses on benchmark problems. While these results are promising, it remains unclear whether the proposed method consistently performs well in more complex real-world applications. Therefore, although the method is intended to be used as a black-box sampler, additional empirical validation may be necessary to support this claim.

The proposed approach intentionally removes the Metropolis-Hastings correction and therefore introduces a controlled asymptotic bias. While the bias is bounded through the EEVPD metric in the Gaussian case, the samples produced by the method are not asymptotically exact for general target distributions. In applications where exact sampling is required or where bias must be strictly controlled, this trade-off may limit the method's applicability.

**Strengths And Weaknesses:**

Strengths :

(1) The algorithm can be used as a black-box sampler, which is potentially beneficial for many application examples. Furthermore, a thorough theoretical analysis for Gaussian target distributions is presented in the paper.

(2) The proposed structure seems to be universal and may be used for any Monte Carlo technique that has an energy error concept.

Weaknesses:

(1) The theoretical guarantees are derived only for Gaussian target distributions, even though the paper empirically demonstrates that the suggested relationship holds for a variety of non-Gaussian examples. Non-Gaussian targets are a common feature of real-world problems, and it can be challenging to numerically confirm the bias bound for every new application, as stated in the paper. Therefore, it would be beneficial if the authors could provide more detail about situations where the approach might become less dependable or suboptimal for non-Gaussian distributions.

(2) In connection with the previous point, it might be challenging to determine whether the asymptotic bias is still small enough in practice when the covariance matrix bound cannot be numerically verified. The method's practical applicability would be enhanced by further discussion on how users should identify or manage the bias in such situations.

(3) The proposed method relies on choosing a target EEVPD value, which defines the step size. It is unclear how sensitive the performance is to this parameter in practice, even though the paper offers some heuristic guidance for selecting this value. It would be beneficial if the authors could discuss how sensitive the step-size adaptation is to this approximation and whether the final step size could be impacted by the inaccurate EEVPD estimation in early burn-in iterations.

(4) The proposed method may serve as an effective black-box sampler. Nonetheless, benchmark problems are the primary focus of the experiments. The claim that the method could be seen asa general-purpose black-box would be strengthened by additional experiments on more practical large-scale example (e.g., high-dimensional datasets or modern machine learning models like logistic regression or neural network models on datasets like MNIST or CIFAR10).

---

> ### Author Rebuttal · Authors · 2026-03-30
>
> We would like to thank the reviewer for the useful comments and suggestions. We have incorporated them in the text which we believe significantly improved the paper, both by adding new experiments and by clarifying means to identify large discretization error cases.
>
> ## 1. and 2. : Identifying problematic cases
> Indeed the provided Gaussian bounds do not always hold in general (even though they often do, as shown in the numerical experiments).
>
> The situation should be compared to the issue of initialization bias in adjusted methods, where there is no way to prove this bias has vanished, and instead, diagnostics must be used after running the chain. One such example is Gelman-Rubin $\widehat{R}$ which is commonly used to diagnose unacceptable dependence on the initialization, by running independent chains with a different initialization and comparing their expectation values. If the diagnostic fails one can run longer chains and recalculate the $\widehat{R}$, but there is no theoretical guidance that would enable one to choose a chain length in advance and have a guarantee that the initialization bias is below a predefined threshold.
>
> Analogously, for the unadjusted methods, we propose (currently in the limitations section, but we will move this to a more prominent location) to run diagnostics which can identify that the discretization bias is non-negligible. The idea is to additionally run two independent chains with half the step size and compare the expectation values obtained by their combined samples with the expectation values obtained by the samples from the original chain. Since the discretization bias typically strongly depends on the step size, this test can identify if the discretization bias affects the expectation values at an unacceptable level.
> If this is the case, we must run a chain with a shorter step size.
>
> ## 3. Sensitivity to the EEVPD
> Ablation study with respect to the EEVPD was performed in Appendix E, see Figure 7. It shows that the algorithm converges to the low error ($b^2 < 0.01$) for a broad range of EEVPD values and the speed of convergence is also not very sensitive to the EEVPD. Stepsize is continuously adapted throughout the burn-in such that the initial samples are quickly forgotten. The rate of forgetting is set by the parameter $\gamma$ which we fix to ensure a decay time of 50 steps. This is typically short compared to the length of the burn-in, thus the initial samples do not affect the final stepsize adaptation. We have verified that the stationary chain (chain that discards a very long burn-in) that uses the stepsize obtained by the proposed adaptation algorithm indeed has an EEVPD that the adaptation algorithm was targeting. This demonstrates that the adaptation was not compromised by the burn-in samples.
>
> ## 4. Experiment
> Following the suggestion, we have now added the quantum U(1) gauge theory example in two Euclidean dimensions (see e.g. [1]) on a $L \times L$ lattice with the periodic boundary conditions. We set the inverse temperature $\beta = 2$. For a million dimensional example ($L = 1024$), adjusted HMC converges to low error on the Polyakov loop observable (a physically relevant expectation) in 17,699 steps and adjusted MCLMC in 15,578 steps, while unadjusted LMC converges in 102 steps and unadjusted MCLMC in 198 steps. This striking improvement originates from the flat scaling of ESS/grad with dimension in the unadjusted case.
>
> We have avoided pure ML applications as it is not yet clear whether or not the Bayesian neural networks offer any advantage over the standard neural networks. We are however applying the proposed method to scientific applications, namely cosmology and quantum field theory, but the complexity of these problems sets them outside of the scope for this paper.
>
> We would also like to point out that the $\phi^4$ physics model is a practical large scale (million parameters) example, which we explored in the paper. Studying statistical field theories like the $\phi^4$ model is an active field of research. Although often used as a benchmark, the stochastic volatility model uses real data and is high dimensional (> 2000 parameters).
>
> [1] Kanwar, Gurtej, et al. "Equivariant flow-based sampling for lattice gauge theory." Physical Review Letters 125.12 (2020): 121601.
>
> ## Limitations
> We’d like to ask the reviewer to clarify when the exact samples are needed in practice. We are not aware of any such problems, since the Monte Carlo error always also contains variance caused by the finite number of samples. Put differently, as long as the bias of the samples is sufficiently small it is not possible to distinguish them from the exact samples given only a finite many samples.

---

> > ### Author Rebuttal · Reviewer_4282 · 2026-04-03
> >
> > Thank you for the thoughtful rebuttal. The added diagnostic for identifying problematic discretization-bias cases, the clarification on EEVPD sensitivity, and the new large-scale experiment substantially address my main concerns. I still view the Gaussian-only nature of the formal guarantee as a limitation, but the rebuttal makes a convincing case that the method is practically useful beyond that setting. I therefore raise my recommendation to weak accept.

---

### Official Review · Reviewer_tSw1 · 2026-03-12

**Soundness:** 3
**Presentation:** 4
**Significance:** 3
**Originality:** 4
**Overall Recommendation:** 6
**Confidence:** 3

**Summary:**

The authors explore gradient-based MCMC samplers that are unadjusted i.e. do not have a Metropolis Hastings accept/reject step. They explain this tradeoff as trading bias for variance, and claim that for a small enough step size the bias is negligible relative to the unavoidable variance. To find such a step size, they pick a step size that controls the energy error variance per dimension (EEVPD) for a Gaussian distribution. They then perform convincing experiments on both Gaussian and non-Gaussian Bayesian inference problems.

**Compliance With Llm Reviewing Policy:**

Affirmed.

**Final Justification:**

This paper presents a compelling case for unadjusted (Metropolis-free) gradient-based MCMC samplers, with an automatic step size selection method that controls the energy error variance per dimension (EEVPD) calibrated on Gaussian targets.

**Soundness.** The core insight — that unadjusted samplers trade bias for variance, and that the bias can be made negligible relative to irreducible variance via principled step size control — is well-argued. The theory is developed for Gaussian distributions and applied to non-Gaussian targets, which is a standard approach in the MCMC literature. The rebuttal provided useful intuition for why the Gaussian-calibrated bound transfers: the energy error is typically more sensitive to step size increases than the asymptotic bias in non-Gaussian settings, so the Gaussian bound remains conservative. The extensive non-Gaussian experiments corroborate this empirically.

**Originality.** The fundamental insight is original and the EEVPD-based step size tuning is a practical contribution. The resulting method is simple, automatic, and outperforms optimally tuned alternatives including NUTS — a high bar.

**Significance.** Removing the Metropolis test while maintaining practical accuracy is valuable for scalability. The method's strong performance across both Gaussian and non-Gaussian Bayesian inference problems, combined with its simplicity, makes it likely to see adoption.

**Clarity.** The paper is exceptionally well-written — one of the clearest expositions I have reviewed. The experimental setup and comparisons are thorough and transparent.

The rebuttal addressed both questions satisfactorily — the authors clarified the relationship between generalized HMC and underdamped LMC, and provided helpful reasoning on Gaussian-to-non-Gaussian transfer. I maintain my score of 6 (Strong Accept).

**Key Questions For Authors:**

1. In the related works section, can you explain how Generalized Hamiltonian Monte Carlo [1] relates to the other methods discussed, if relevant?
2. Like many MCMC papers, theory is derived for a Gaussian distribution and then extended to non-Gaussian distributions. The authors include many convincing experiments on non-Gaussian distributions to demonstrate that their theory works. However, do the authors have any additional guidance or reasoning on how/why theoretical results for Gaussians translate to non-Gaussian settings? Any insights would be helpful.

[1] Horowitz, Alan M. "A generalized guided Monte Carlo algorithm." Physics Letters B 268.2 (1991): 247-252.

**Limitations:**

Yes

**Strengths And Weaknesses:**

Strengths:
1. The fundamental insight of the paper is original and compelling: unadjusted MCMC samplers trade bias for variance, but the bias can be controlled by clever step size selection
2. Experiments are very strong, outperforming many other optimally tuned methods on both Gaussian and non-Gaussian targets, including the NUTS sampler
3. The paper is written incredibly clearly

Weaknesses:
1. Convincing experiments show that their automatic step size tuning method works on both Gaussian and non-Gaussian distributions. Nonetheless, the theory to control step size is developed for Gaussian distributions, but then applied to non-Gaussian distributions. This is a minor concern and does not impact my score.

---

> ### Author Rebuttal · Authors · 2026-03-30
>
> We thank the reviewer for the useful comments and suggestions and address the key questions below:
>
> 1. We refer to the generalized HMC as to underdamped Langevin Monte Carlo (LMC), as it is a discretization of the underdamped Langevin dynamics. We discuss LMC in the related work and cite the suggested reference in line 35 (right column). We will clarify that generalized HMC and LMC are the same algorithms in the related work section to avoid confusion.
>
> 2. Increasing the step size of the unadjusted method increases both the energy error and the asymptotic bias. The observation that the bound seems to typically hold for non-Gaussian distributions suggests that the energy error is more affected by the step size increase than the asymptotic bias, as compared to the standard Gaussian distribution (for which the bound is an equality). For example, we have shown that having different scales in the posterior (non-standard Gaussian target) affects the energy error more than it does the bias. Possibly something similar happens in many practical non-Gaussian examples as well.

---

> > ### Author Rebuttal · Reviewer_tSw1 · 2026-04-01
> >
> > Our concerns are fully resolved by the rebuttal. We maintain our score.

---

### Decision · Program_Chairs · 2026-04-30

**Decision:**

Accept (regular)

**Comment:**

The paper is strong overall: it tackles an important and practically relevant problem in MCMC by making unadjusted HMC/LMC methods usable through an automatic step-size tuning rule, and the reviewers broadly agree that the core idea is original, technically meaningful, and supported by strong empirical results. The main residual weakness is that the formal guarantees are limited to Gaussian targets, while the broader claims rely on empirical evidence and diagnostics in non-Gaussian settings. Still, the rebuttal appears to have addressed the reviewers’ main concerns well: all reviewers ended at accept or strong accept, and several explicitly raised their scores after the response. Given the combination of theoretical contribution, practical impact, and convincing evaluation, I view this as a clear accept.